# The effects of death and post-mortem cold ischemia on human tissue transcriptomes

Pedro G. Ferreira *et al.*[#] (ID)

Post-mortem tissues samples are a key resource for investigating patterns of gene expression. However, the processes triggered by death and the post-mortem interval (PMI) can significantly alter physiologically normal RNA levels. We investigate the impact of PMI on gene expression using data from multiple tissues of post-mortem donors obtained from the GTEx project. We find that many genes change expression over relatively short PMIs in a tissue-specific manner, but this potentially confounding effect in a biological analysis can be minimized by taking into account appropriate covariates. By comparing ante- and post-mortem blood samples, we identify the cascade of transcriptional events triggered by death of the organism. These events do not appear to simply reflect stochastic variation resulting from mRNA degradation, but active and ongoing regulation of transcription. Finally, we develop a model to predict the time since death from the analysis of the transcriptome of a few readily accessible tissues.

---

Post-mortem human tissue samples are a valuable resource for biological research. Specifically, use of post-mortem material is crucial for studying the patterns of normal gene expression underlying tissue specificity within individuals, as sampling such tissues from living individuals would be impossible. However, the death of an organism triggers a cascade of events that ultimately, in a relatively short time frame, lead to cell death and autolysis. Although DNA is known to be relatively stable over long post-mortem periods, RNA is much more labile in nature, and sensitive to degradation in a tissue-specific manner[1]. There are conflicting reports on how the post-mortem interval affects RNA integrity[2–10] but several studies, in different mammals, have shown that RNA can remain largely intact even for considerable time periods, when samples remain properly stored. In addition, a variety of pre-mortem factors, including environmental parameters and the circumstances of death, may also influence the quality of the collected tissues and their RNA[8,11]. RNA quality impacts measures of gene expression. Recent studies[12–15] have shown that sequencing lower RNA quality samples, as measured by the RNA integrity index (RIN)[16], leads to a decrease in the quality of the data obtained by high throughput RNA sequencing (RNA-seq), and the use of RIN, and other related variables, as covariates in differential expression analysis, has been recommended[12,13,17].

On the other hand, transcriptional changes are expected to occur as a response to the death of an organism. However, little is currently known about how death and the length of the post-mortem cold ischemia interval specifically affect gene expression since most existing reports are based on very few genes, tissues or individuals[5–7,10,11,17,18]. Therefore, RNA levels measured in post-mortem tissue samples will be affected both by biological responses to organism death, as well as to RNA degradation occurring as a consequence of cell death. Understanding how these effects are dependent on the post-mortem interval is essential for the proper use of post-mortem gene expression measures as a proxy for ante-mortem physiological gene expression levels[5,10,18–20].

Here we analyze the GTEx[21–25] RNA-sequencing data to investigate the impact of death and the post-mortem cold ischemic interval on the transcriptomes of human tissues. We find that different tissues have a different response over the time elapsed since death, but that when appropriate covariates are identified and taken into account, the impact of death on tissue transcriptomes can largely be controlled. We identify the cascade of molecular events triggered by death specifically in the Blood transcriptome. Finally, we develop a model to predict the time since death from the analysis of the transcriptome of a few readily accessible tissues.

## Results

**Study overview.** We used mRNA sequencing data from the GTEx project (V6, Supplementary Table 1 and 2), and the derived gene and transcript quantifications obtained on Gencode[26] V19. We restricted our analyses to 36 tissues with >20 samples, including whole blood and two brain sub-regions (cortex and cerebellum) for a total of 7105 samples, corresponding to 540 donors (Supplementary Fig. 1, 2, 3, Methods). All samples were collected and preserved with the PAXgene Tissue preservation system[21].

The GTEx metadata contains an extensive annotation of samples and donors, including the postmortem interval (PMI). For GTEx individuals, PMI is defined as the time since death to the start of the GTEx collection procedure. For tissue samples, this is defined as the time in minutes spanning the window from the moment of death, or the cessation of blood flow, until tissue stabilization and/or preservation takes place, with values ranging from 17 to 1739 min (Fig. 1a, Supplementary Note 1). Correlation analysis shows that there is a strong association of PMI with variables describing tissue recovery and death circumstances, as these variables are correlated and reflect the same intrinsic features of the collection procedures (Supplementary Fig. 4, Supplementary Table 3). The relationship between PMI and RNA stability is very tissue-dependent (Fig. 1b, Supplementary Fig. 5, Supplementary Table 4), in agreement with previous observations[5,17,27].

**Impact of PMI on gene expression.** To identify genes that changed expression depending on PMI, we used the five PMI intervals also used by the GTEx Biospecimen Methodological Study (BMS)[21], and asked which genes had a significant and noticeable change between two consecutive time intervals (>2-fold change and Wilcoxon test $p < 0.05$, see Methods and Supplementary Note 2). The number of genes with a significant change in at least one interval transition varies widely between

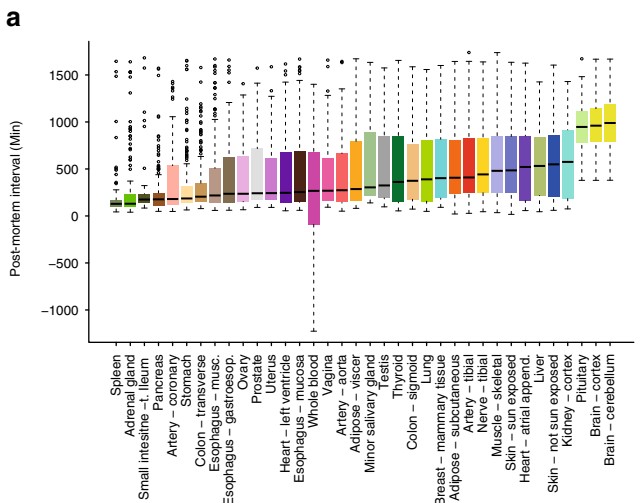

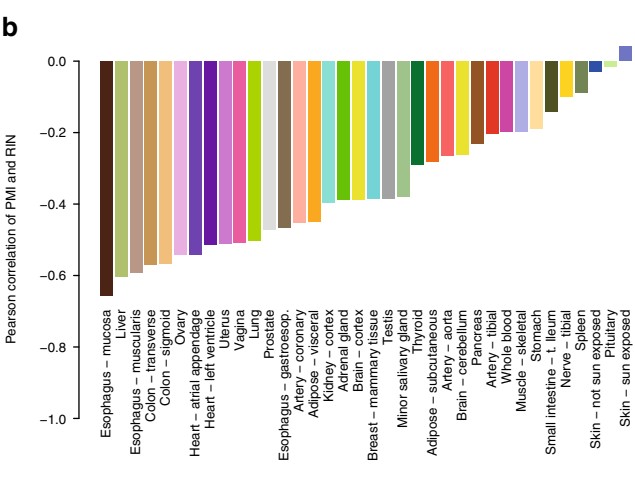

**Fig. 1** Characteristics of the samples and tissues used in this study. **a** Distribution of PMI values (in minutes) with tissues ordered by the median value. Whole blood contains samples with negative time corresponding to samples obtained pre-mortem. **b** Distribution of Pearson correlation between PMI and RIN values. Esophagus, Liver, Colon, Ovary, Uterus, Vagina, and Heart are the tissues in which RIN is more affected by PMI ($r < -0.5$), while Skin, Pituitary, Spleen and Nerve are the ones in which is less affected ($r > -0.1$).

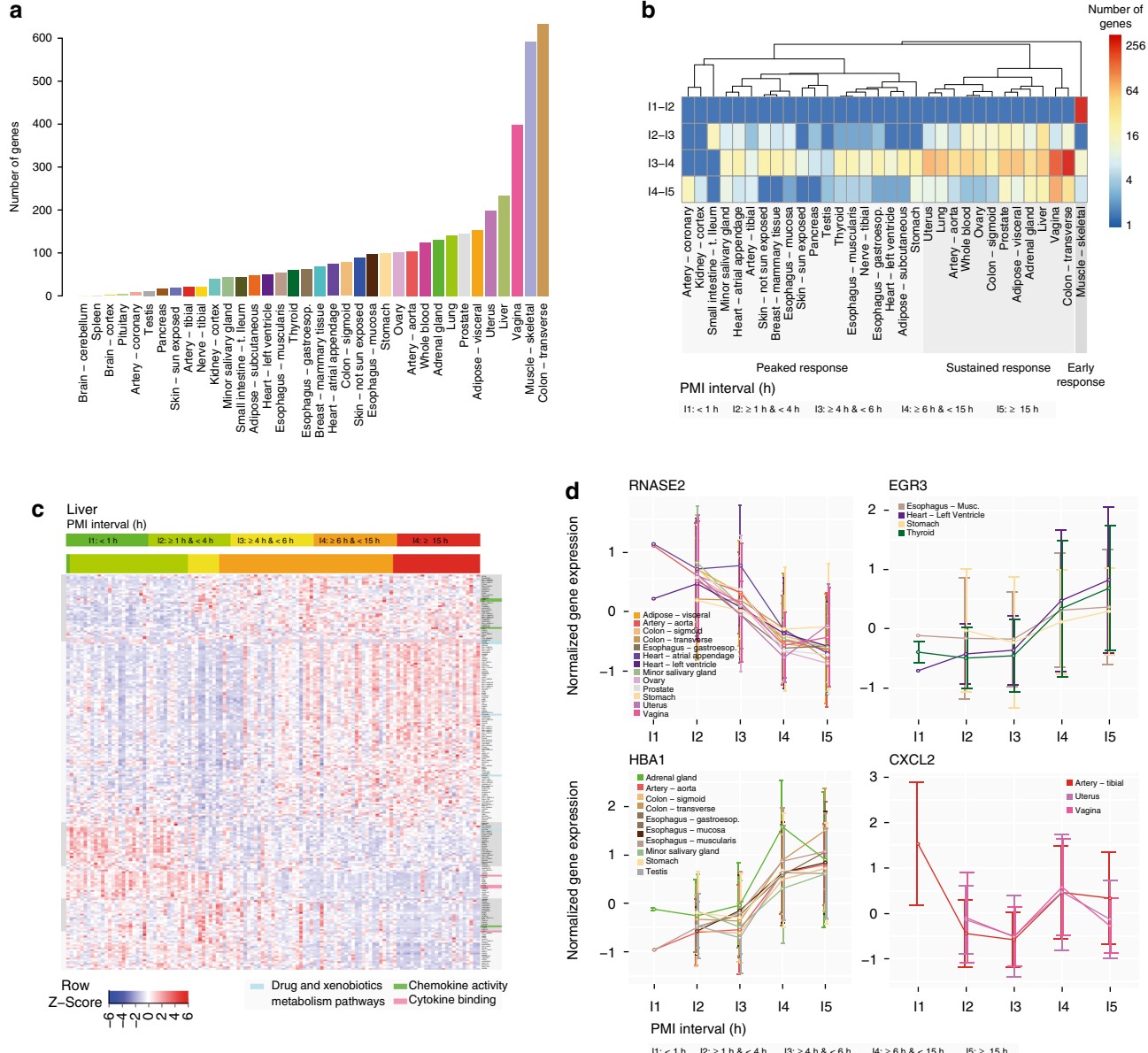

**Fig. 2** Effect of PMI on gene expression. **a** Distribution of the number of genes with significant temporal changes per tissue between at least two time intervals. Brain and pituitary have longer PMIs, only within last two time intervals, thus less interval ranges to detect significant changes. **b** Heatmap with the number of genes with significant changes detected between two consecutive time intervals. **c** Heatmap with normalized expression values for genes with changes in liver. The top bar is the color code for the PMI interval of each sample, the right bar list genes involved in the various functions and pathways. On the side we highlight sub-clusters with different patterns of temporal expression. **d** Example of four genes with different temporal patterns. *RNASE2* is a non-secretory ribonuclease involved in several functions; *HBA1* is alpha hemoglobin involved in oxygen transport; *EGR3* is a transcriptional regulator involved in early growth response; *CXCL2* is a chemokine gene that encodes secreted proteins involved in immune and inflammatory processes

tissues, ranging from none in brain cerebellum and spleen, to >600 in muscle and colon transverse (Fig. 2a). Although most tissues are characterized by a sharp shift in gene expression at around 6 h after death, there are remarkable differences between tissues regarding the transcriptional response to PMI (Fig. 2b). Some tissues (e.g., muscle) exhibit an early response, with most genes that change expression doing so right after death (Supplementary Fig. 6). Another set of tissues show a more sustained response, with gene expression changes of similar magnitude occurring through all PMI intervals (Fig. 2c). Finally, another set of tissues show a peaked response, with most changes occurring between the intervals of 4–6 h and 6–15 h (Supplementary Fig. 7).

There is little overlap of affected genes across the tissues. We identified 187 genes (94 are protein-coding) with post-mortem

gene expression changes in at least three tissues (Supplementary Fig. 8). The gene that showed consistent changes across the largest number of tissues was *RNASE2*, a gene from the family of ribonucleases, enzymes involved in the degradation of RNA. *RNASE2* shows a consistent decrease in expression across 13 tissues (Fig. 2d). Two alpha globin genes, *HBA1* and *HBA2*, involved in the transport of oxygen from the lung to the peripheral tissues, show an increased expression in several tissues but not in blood, where they are the most expressed genes (Fig. 2d). Several histone genes show increased patterns of expression in line with previous results[28,29] (Supplementary Fig. 8, Supplementary Data 1). Growth factors, such as *EGR3* also have an increased expression from 4hr to later on (Fig. 2d, Supplementary Fig. 8). Other genes such as the chemokine

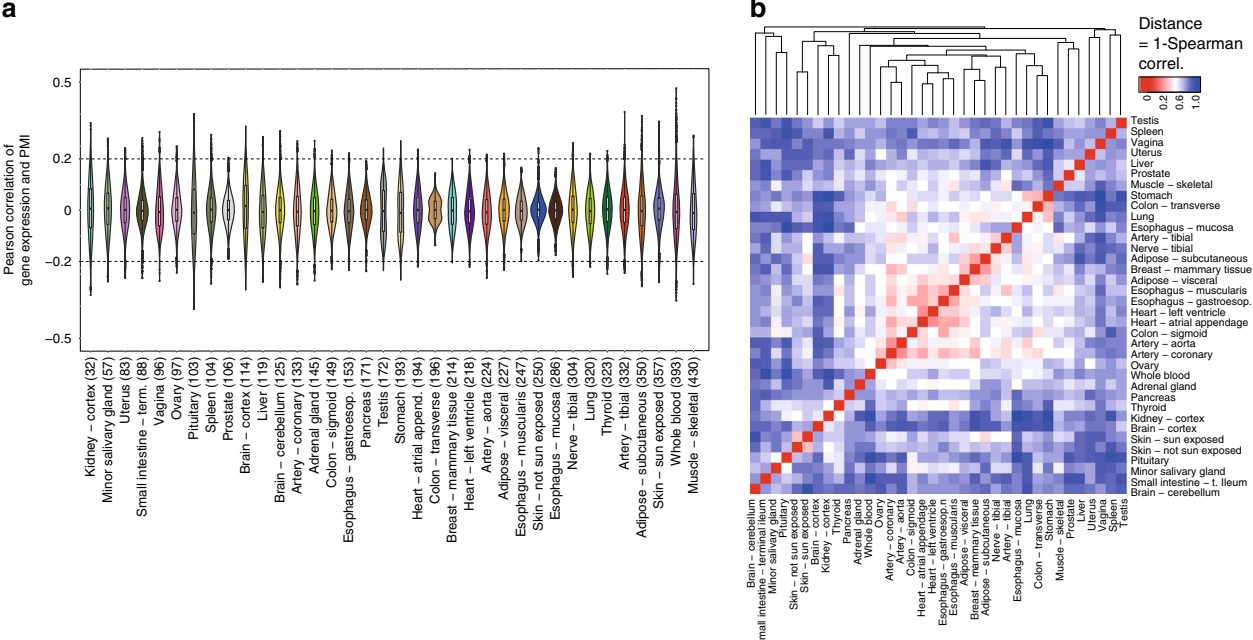

**Fig. 3** PMI and gene expression correlation patterns. **a** Distribution of Pearson correlation between gene expression and PMI, across the different tissues (sorted by sample size, in parenthesis). Only for a few genes, this correlation exceeds an absolute r-value of 0.2. **b** Clustering based on the ranking (Spearman) correlation of the values in (**a**) show that sub-tissues of a given tissue or closely related organs have the similar patterns of correlation

*CXCL2* show a more dynamic behavior with expression changes in opposite directions at subsequent intervals (Fig. 2d). Gene ontology analysis of the genes affected across several tissues (Supplementary Fig. 8) shows enrichment for genes in the extracellular region and genes involved in nucleosome and chromatin assembly and in protein–DNA complexes. There is also enrichment for inflammatory and immune response processes.

While there are noticeable changes in gene expression associated with PMI, we nonetheless found that the characteristic transcriptional signature of tissues remains largely intact through the PMI intervals considered here. We clustered the GTEx samples at these intervals and measured, using modularity (see Methods), how well the clustering recapitulates tissue type. Here, we compute modularity on the network constructed from gene expression correlations between samples when the data are grouped by tissues. Modularity remained stable through the PMI intervals at any threshold of the correlation defining the network edges (see Supplementary Fig. 9).

Because PMI dependent expression changes are largely tissue-specific, they could confound tissue differential gene expression since the observed effects could be caused by differential response to PMI rather than by differences in tissue biology. To investigate to what extent these effects can be controlled for, we used a linear regression model that allows incorporating additional covariates. We specifically selected fourteen variables, predominantly demographic, medical history and sample QC metrics that are orthogonal to the sample collection procedure, to include as expression covariates in the model[21] (see Supplementary Fig. 4, Supplementary Table 5 and 2). These are essentially the covariates employed in the GTEx eQTL analyses[24]. Residuals were then used as the expression phenotype and the Pearson correlation (r) as a measure of linear relationship with PMI (Methods, Supplementary Notes 2). On average we found only 54 genes per tissue (0.2%), which showed significant correlation of gene expression with PMI (FDR < 1%) (Fig. 3a, Supplementary Table 6), compared to 6919 genes per tissue (39.3%), if using the same

model without covariates. In most of these cases, however, the effect is small (only 189 (1.1%) with $r < |0.2|$ (Supplementary Fig. 10). Moreover, clustering of tissues based on the ranking of correlations gene expression-PMI generally recapitulates tissue type (Fig. 3b). These results suggest that the effect of PMI on measured gene expression is relatively modest and can be further minimized by using appropriate covariate correction in analyses. The effect is weakly mediated by the number of exons, the length of the gene and of the coding region and GC content (Supplementary Table 7). PMI has also little effect on the proportion of intergenic RNA-seq reads, as well as on 3' mapping bias, commonly observed in RNA degraded samples[12,14,30–32] (Supplementary Figs. 11–15, Methods).

To specifically analyze the impact of PMI in energy metabolism, we investigated its relationship with mitochondrial RNA (mtRNA) levels. We observed no significant changes in mtRNA concentration across different RIN values, and donor ages (Supplementary Figs. 16 and 17). Across most tissues, samples exhibit a significantly lower proportion of mitochondrial reads in late PMIs (Fig. 4a, Methods), except blood, salivary gland, heart-left ventricle and, particularly, liver that exhibits a substantial higher proportion of mitochondrial RNAs for late PMIs (Fig. 4b, Supplementary Fig. 17). Decreasing mtRNA abundance across all PMI intervals is observed specifically in female tissues (ovary, vagina, and uterus, see Fig. 4c).

Finally, we investigated the effect of PMI on splicing. We calculated the inclusion levels[33] of internal exons (Supplementary Fig. 18a, Methods). We then performed linear regression analysis of PMI and PSI values and found 1,399 exons (612 unique) significantly correlated with PMI ($|r| > 0.5$ and FDR ≤1%; Fig. 5a–c), of which 160 were observed in three or more tissues (Fig. 5a). In contrast to gene expression, there is a substantial sharing of exons among the top affected tissues (those with ≥ 20 significant exons), with the tissue pairwise overlap ranging from 43% to 82%, representing 22 to 76 shared exons. Functional analysis of genes with recurrent exons (i.e., with association with PMI in more than two tissues) shows a noteworthy enrichment

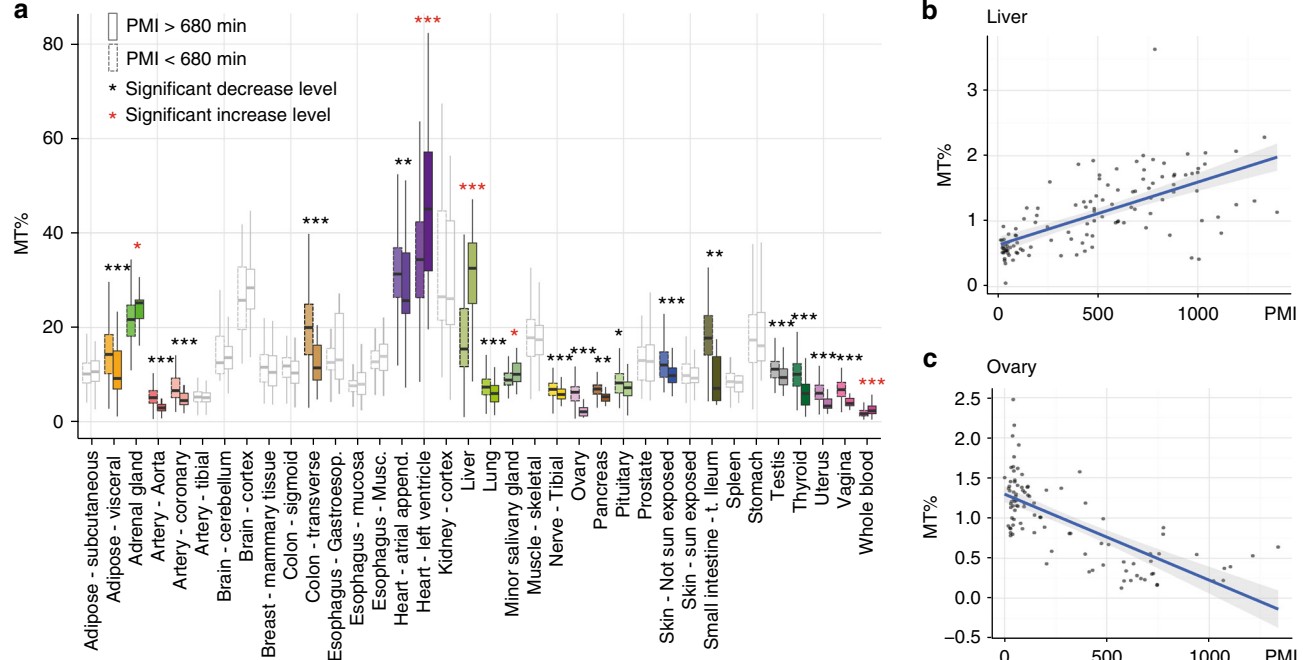

**Fig. 4** Effect of PMI on mitochondrial transcription and splicing. **a** Proportion of RNA-seq reads originating from mitochondrial genes (mtRNA concentration) in early (≤680) and late (>680) PMI intervals. **b**, **c** mtRNA concentration depending on PMI in Liver (**b**) and Ovary (**c**)

on RNA binding and RNA splicing genes (Supplementary Fig. 18b, c). We also investigated if, as a consequence of death, we could observe a generic alteration of splicing. As a proxy for splicing alteration, we computed the Shannon's entropy on the relative abundance of a gene's alternative splicing isoforms (Methods)—higher values corresponding to more stochastic production of alternative isoforms. We did observe an increase of splicing entropy in many cases (Fig. 5d, e), although not a systematic trend across all tissues (Supplementary Fig. 19).

**Changes induced by death in the whole blood transcriptome.** Among the samples collected for GTEx, the blood samples are unique in having been collected pre-mortem for some donors and post-mortem for others. This provides an opportunity to assess the impact of death on the gene expression of a specific tissue. Dimensionality reduction (MDS) and hierarchical clustering of gene expression profiles clearly distinguishes pre- and post-mortem states of blood samples (Fig. 6a and Supplementary Fig. 20). The "cause of death" (assessed by the 4-point Hardy scale classification, Supplementary Notes 1) is quite different for individuals from whom Blood was obtained pre-mortem compared to post-mortem, but this does not appear to have a major impact on the clustering, which is independent of Hardy classification (see Methods, Fig. 6a).

To characterize changes in gene expression that are triggered by death, we identified genes that were differentially expressed between pre-mortem and post-mortem blood samples, the latter being collected at several different PMI intervals (Fig. 6b, Supplementary Table 8 and Supplementary Data 2). Immediately following death (and up to seven consecutive hours) we observe an increase in the expression of many genes, and a decrease in the expression of a few. The majority of the changes in gene expression, however, occur between 7 and 14 h post-death, with thousands of genes showing differential expression (equally in both directions) relative to pre-mortem samples. Then, between 14 and 24 h, the transcriptome seems to stabilize, with comparatively few genes showing differential expression relative to pre-mortem samples (among those that do, there are more

over-expressed than under-expressed). Categorizing the nature of these changes in gene expression in blood samples following death, we observed five main functional activities[34] (Fig. 6c, Supplementary Table 9, Supplementary Notes 3): 1) changes in DNA synthesis and fibrinolysis; 2) deactivation of the immune response; 3) an increase in activity of processes related to cell necrosis; 4) an abrupt inactivation of carbohydrate metabolism, synthesis of lipids (e.g., cholesterol) and ion transport; and 5) an activation of processes related to Blood coagulation and Response to stress (Supplementary Fig. 21a). Specifically, the way in which carbohydrate metabolism is affected, with severe deactivation of the tricarboxylic acid cycle, while glycolysis is activated (FDR < $10^{-27}$; Fig. 6d, Supplementary Table 9), suggests that hypoxia is likely playing a major role in the initial pre- to post-mortem transition (FDR $7.2 \times 10^{-67}$). More gradually, the immune system is also deactivated (several immunity-related functions with FDR < $10^{-30}$, Supplementary Table 9, Supplementary Fig. 21b). In addition, a response to stress, along with the detection of DNA damage and the activation of the corresponding repair machinery is observed (FDR < $10^{-14}$; Supplementary Table 9). Finally, a general arrest of cell proliferative processes occurs. Processes like growth arrest are activated and others, like Initiation factor, the starting process of protein production, are dramatically deactivated.

The transcriptional changes detected above may partially be related to changes in the cellular composition of blood triggered by death. Indeed, blood is a complex tissue composed of multiple cell types. We investigated differences in cell composition between blood samples collected pre- and post-mortem. We used CIBERSORT[35], to deconvolute bulk gene expression into expression levels for 18 different cell types. We found significant differences in overall cellular composition between pre- and post-mortem blood samples ($p < 0.001$), the most notable changes induced by death being an increase in resting NK cells and CD8 T-cells, and a substantial reduction in neutrophils (Fig. 7a). These results are consistent with the observed deactivation of the immune system (Supplementary Fig. 21b), since similar trends are observed to be associated with dysregulation of the immune

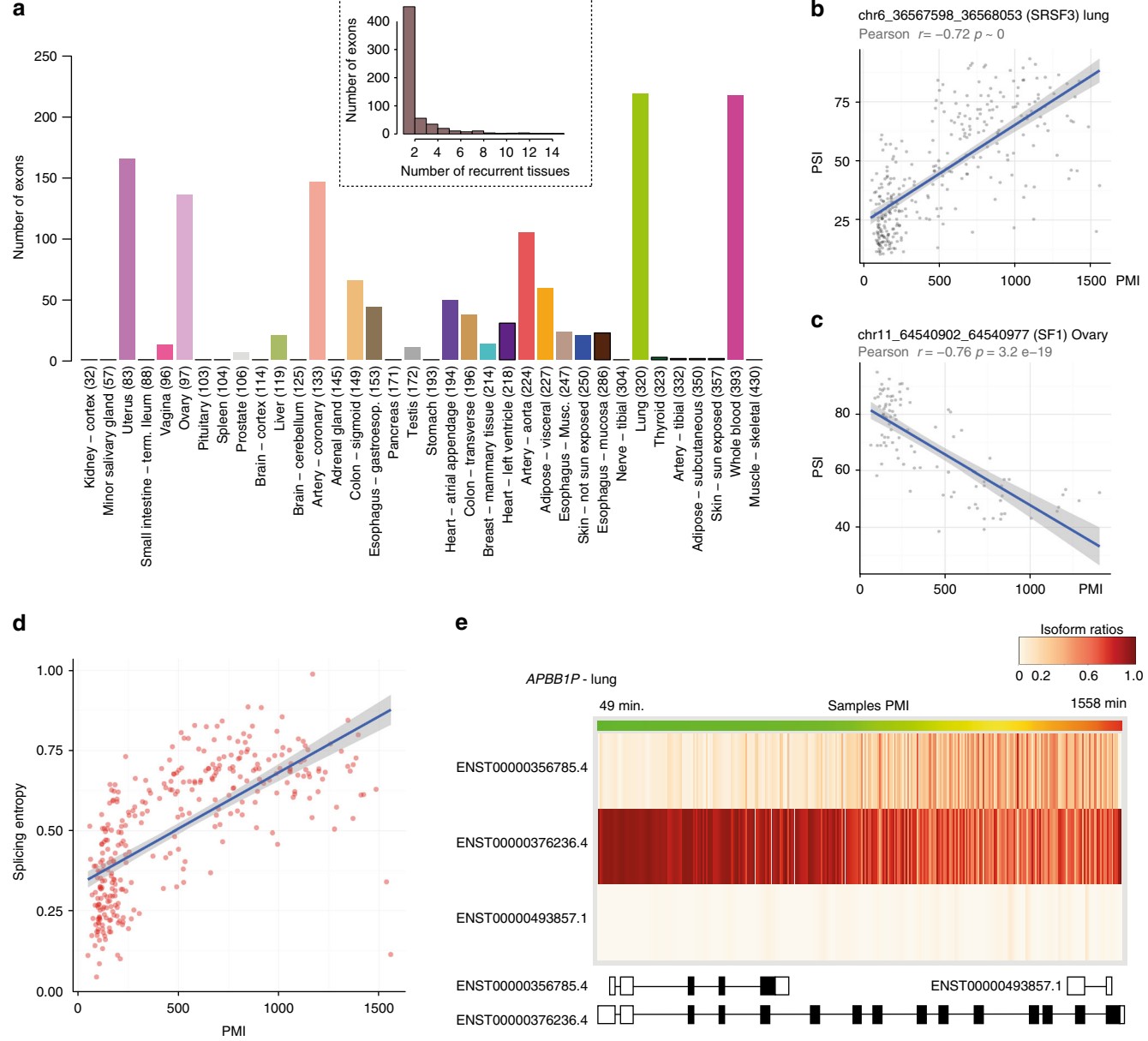

**Fig. 5** Effect of PMI on splicing. **a** Distribution of the number of exons with significant differential inclusion across the different tissues; inset: number of exons with differential inclusion occurring in multiple tissues. **b**, **c** Examples of two exons with PMI-correlated inclusion levels. *SRSF3* and *SF1* encode pre-mRNA splicing factors that form part of the spliceosome. **d** Splicing Entropy for the *APBB1IP* gene depending PMI in the lung tissue. **e** *APBB1IP* gene has three isoforms. The heatmap shows the proportion of each isoform in the Lung samples sorted by PMI (1 indicates that only one isoform contributes to the expression of the gene and 0 that the isoform is not expressed) PMI of the samples range from 49 to 1558 minand are represented in the green to red gradient scale bar. Exon structure (not to scale) for the three isoforms is represented below. This figure depicts how the expression of the longer transcript in this gene becomes less dominant as PMI increases, and, as a consequence, the abundance of the different isoforms tends to converge

system with age. Neutrophils, in particular, are the first cells to migrate to pathogenic infected sites, and a decrease in their levels implies impaired ability to traffic into and out sites of infection.

Death also has an observable impact on splicing in the blood transcriptome. We identified 497 exons (from 381 genes) that were differentially included between the pre- and post-mortem samples ($p < 0.01$, |$\Delta$PSI| > 0.1, Supplementary Fig. 22, Supplementary Table 10). This represents 14% of all exons (3441) that were found to be variable across samples (Methods). Most of these 497 exons (75%) tended to be "included" in the pre-mortem samples (and not in the post-mortem samples), suggesting that splicing deregulation is occurring. Indeed we found that post-mortem samples have a higher entropy than pre-mortem samples

(Fig. 7b), reflecting tighter, more controlled, usage of splicing isoforms in the pre-mortem samples. In general, we found there was an increased usage of the major (dominant) isoform in the pre-mortem samples relative to the post-mortem samples (Supplementary Fig. 23).

**Prediction of the post-mortem interval from gene expression.** The precise estimation of PMI is a problem of central importance in forensic pathology. Traditional methods for this task rely on physical modifications observed on the body, including algor, livor, and rigor mortis[36]. However, these approaches may be unreliable or inaccurate[18]. The use of RNA assays as an addition to the forensic tool kit is of growing interest with studies looking

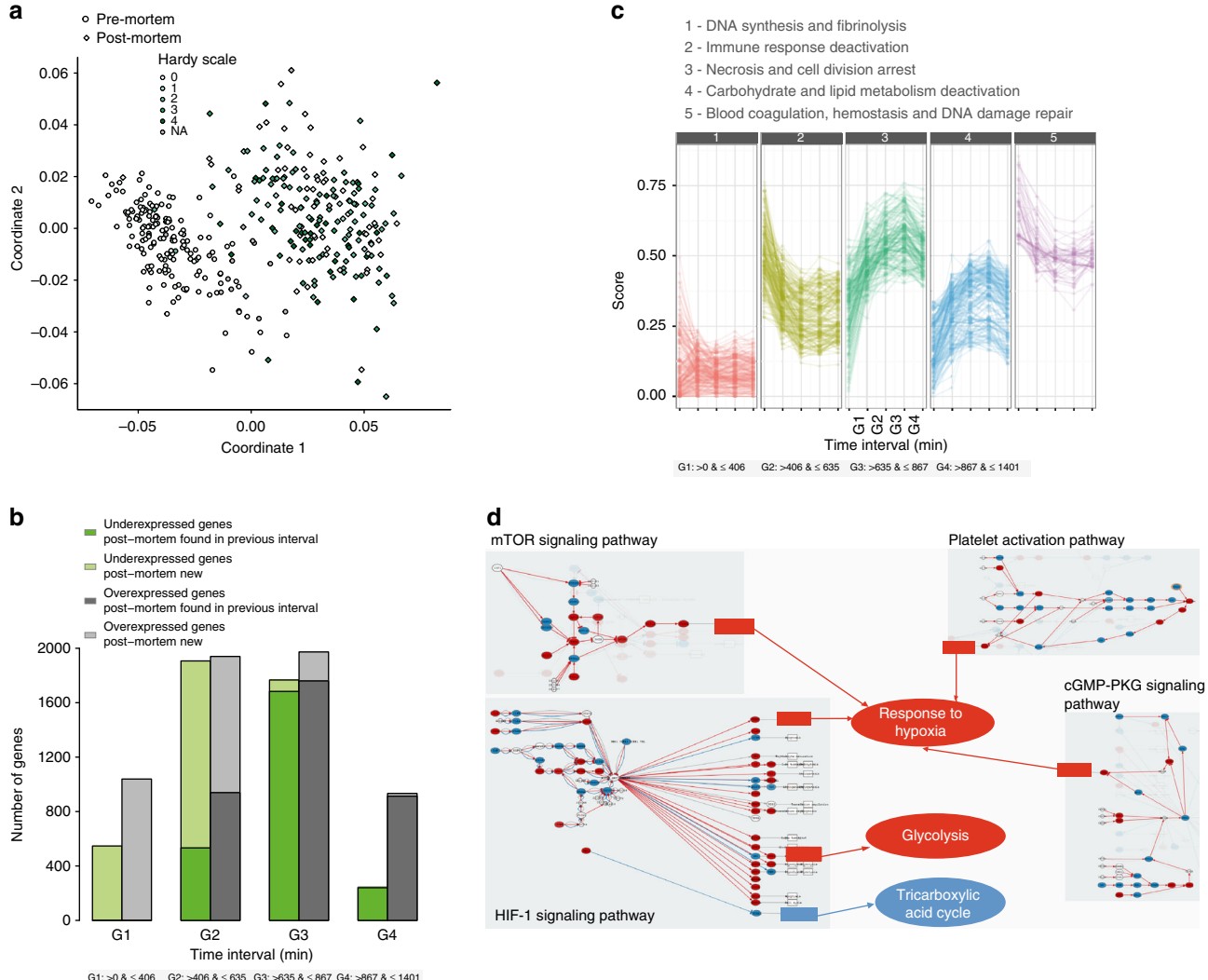

**Fig. 6** Transcriptional changes in blood after death. **a** Multi-Dimensional Scaling of blood samples shows separation between pre and post-mortem samples. Samples are colored by the Hardy scale of the cause of death. **b** Number of genes differentially expressed between the pre-mortem samples and the post-mortem samples stratified at different PMI intervals. Darker filling corresponds to genes that are found as differentially expressed in the previous interval. **c** The five main temporal patterns of change in functional activities upon organismic death. **d** Hypoxia seems to play a major role in the pre-to post-mortem transcription as reflected in the way in which the carbohydrate metabolism is affected (activations in red, deactivations in blue). Response to hipoxia is activated from pathways "Platelet activation pathway" and "cGMP-PKG signaling pathway" through the activation of the corresponding circuits that end in the effector gene ITPR1, annotated as Response to hypoxia, and from pathways "HIF-1 signaling pathway" and "cGMP-PKG signaling pathway" through the activation of the effector gene VEGFA. The "HIF-1 signaling pathway" also activates Glycolysis through the activation of different circuits that trigger effector proteins (PDK1, PFKL, ALDOA, etc.) with annotations such as glycolytic process, canonical glycolysis, glucose metabolic process, etc. The "HIF-1 signaling pathway" also inhibits Tricarboxylic acid cycle through the inhibition of circuits that trigger the effector protein PDHA1 with diverse GO annotations such as tricarboxylic acid cycle, acetyl-CoA biosynthetic process from pyruvate or carbohydrate metabolic process

for a correlation between RNA degradation and PMI[1,10]. The use of mRNA markers in PMI prediction also holds great promise, but so far only a few genes from a handful of human tissues have been tested[18]. Herein, our analyses suggest that the patterns of gene expression change with time after death in a tissue specific manner, and might thus be collectively used to predict the PMI for a given individual. We use the GTEx RNA-seq data to develop and to test such an approach. We first use gradient boosted trees[37] to infer models that use expression of protein coding genes to predict the PMI of each tissue separately. We used data from 399 individuals (about 75% of the 528 available individuals) for training the models and 129 (~25%) for testing (Supplementary Fig. 24 and 25). In the test set we obtained $R^2$ values between predicted and real tissue PMI ranging from 0.78 to 0.16 (Supplementary Fig. 26 and 27), similarly to what was

obtained in the training set (Supplementary Fig. 25). We also calculated for each sample, the difference between real and predicted tissue PMI, and found little deviation on average, although the models tend to overestimate PMI (Fig. 8a). To assess the possibility of overfitting due to the complexity of the data we performed a model stability analysis via resampling (Supplementary Fig. 28). In addition, we used the blood samples and we repeated the training and testing procedures (×100) separately in post-mortem and in pre-mortem samples. We reasoned that if predictions resulted from overfitting, we should be able to predict the time to death in pre-mortem samples equally well as the time since death in the post-mortem samples. Reassuringly, predictions of time to death were essentially random (median $R^2$ 0.02 compared to 0.47 for predictions of time since death, see Supplementary Fig. 29).

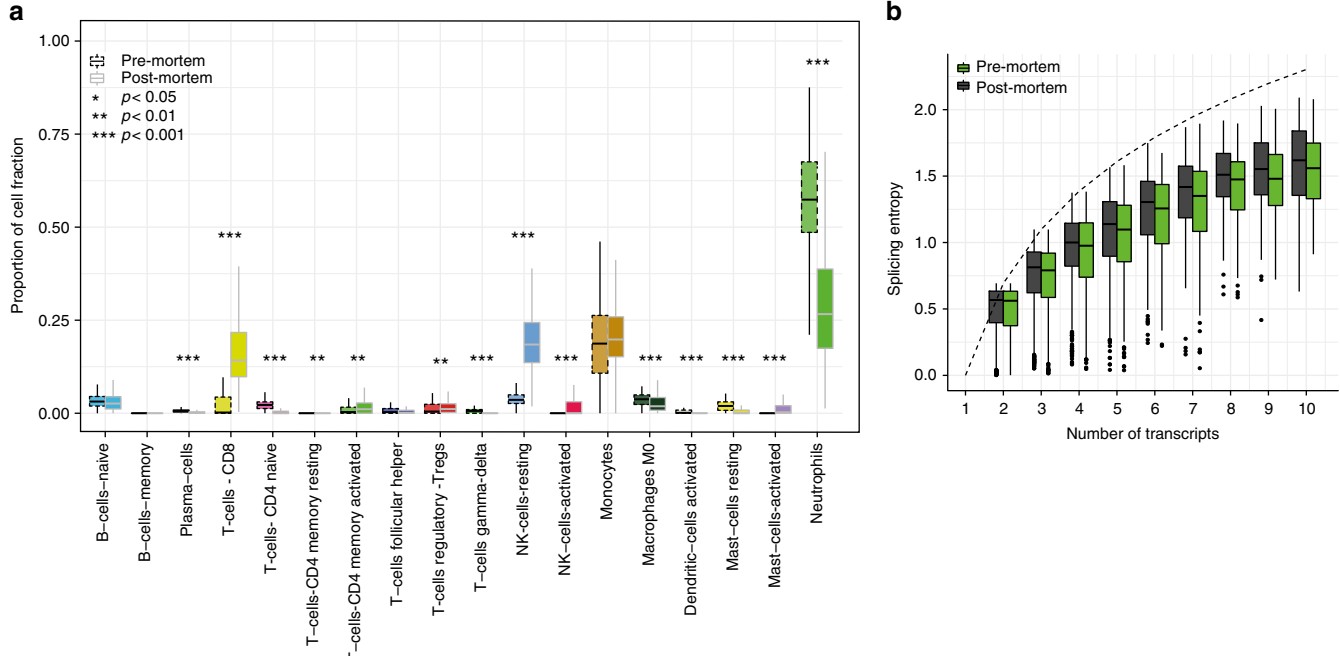

**Fig. 7** Differential cellular composition and splicing entropy in blood. **a** Cellular composition analysis for 18 cell types shows an increase in NK-cells-resting and T-cells-CD8 and decrease in Neutrophils composition from pre- to post-mortem blood samples. **b** Splicing entropy in pre- and post-mortem samples for genes of different number of isoforms

To infer the PMI of each individual, we subtracted from each tissue predicted PMI the time elapsed since the beginning of the GTEx procedure to the processing of that tissue, and averaged the resulting values (Fig. 8c, d). On average, the PMI prediction error (signed difference between real and predicted) is 9.45 min and the median of −63.75 min (Supplementary Fig. 30). The $R^2$ of the predicted and real PMI is 0.77 when all tissues are considered, and 0.8 when using only the top 20 tissues ($R^2 > 0.5$ in the training set, Fig. 8b). As a measure of stability of the PMI prediction on a given individual, we assess the consistency of the tissue PMIs for the individuals. We reason that if all tissues predict consistently very similar PMIs, the prediction of the PMI for the individual is more reliable than if the tissue PMI predictions are very variable across tissues. To assess the consistency of the tissue PMI prediction for a given individual we compute the coefficient of variation (cv), lower values thus indicating more reliable predictions. Supplementary Figure 31 shows the cv distribution on the individuals from the test set.

Since the availability of so many tissues is unrealistic in a forensics scenario, we identified the smallest combination of tissues that can be used to determine an individual's PMI accurately. For each individual in the initial test set, we identified the subset of tissues of a fixed size that can predict the individual PMI with the highest precision. We find that for subsets of sizes 2–6, the tissues that appear more frequently are Adipose—Subcutaneous, Lung, Thyroid, and Skin (Sun Exposed) (Supplementary Fig. 32). We prioritized this approach over simply identifying the combination of tissues with highest $R^2$. Predictions using these four tissues are even superior to those using all top 20 tissues ($R^2 = 0.86$) (Supplementary Fig. 33), and actually only marginally superior to those obtained using some combinations of only two tissues among the four above (Supplementary Fig. 33 and 34).

We investigated to what extent the PMI predictions are robust to the causes of death since this could also have an impact on the transcriptome. To have sample sizes large enough, we grouped the causes of death reported by GTEx (Supplementary Fig. 35a) in

three major death classes: cerebrovascular disease, heart disease, and other causes of death. We did not observe an impact of the class of death in the accuracy of the predictions, as measured by $R^2$ (Supplementary Fig. 35b).

The results above suggest that gene expression values (estimated, for instance, through RNA-Seq) can be used to effectively predict time since death. Figure 9 summarizes the main steps to follow in a putative real case scenario.

We also investigated whether estimates of RNA degradation can be used to predict PMI. We have employed exactly the same methodology, but using the transcript integrity number[15] (TIN) data instead of gene expression. TINs have been proposed to measure RNA integrity based on the uniformity of the read distribution across transcript length[15]. TIN-based predictions of individual PMI have similar accuracy to those based on gene expression (Supplementary Figure 36a and 36b). However, there is only moderate intersection between the two methods on the genes contributing the most to the predictions (Supplementary Figure 36c). This is consistent with our finding that the post-mortem transcriptomic changes are both the result of RNA degradation and of regulated gene expression.

## Discussion

Here we report on the largest systematic study of the impact of death and post-mortem cold ischemia on gene expression across multiple human tissues. Samples obtained post-mortem are a valuable source of material for studies requiring organs and tissues difficult to obtain, or those where it is impossible to study and manipulate them in living organisms. Hence, understanding the impact of death in tissues is essential for the proper interpretation of post-mortem gene expression levels as a proxy for in vivo, living physiological levels.

The death of an organism clearly has an immediate impact on tissue transcriptomes, as illustrated by our analysis of ante- and post-mortem samples. Changes in gene expression as a response to death, and during subsequent post-mortem ischemia, might be expected to reflect stochastic variation resulting from the

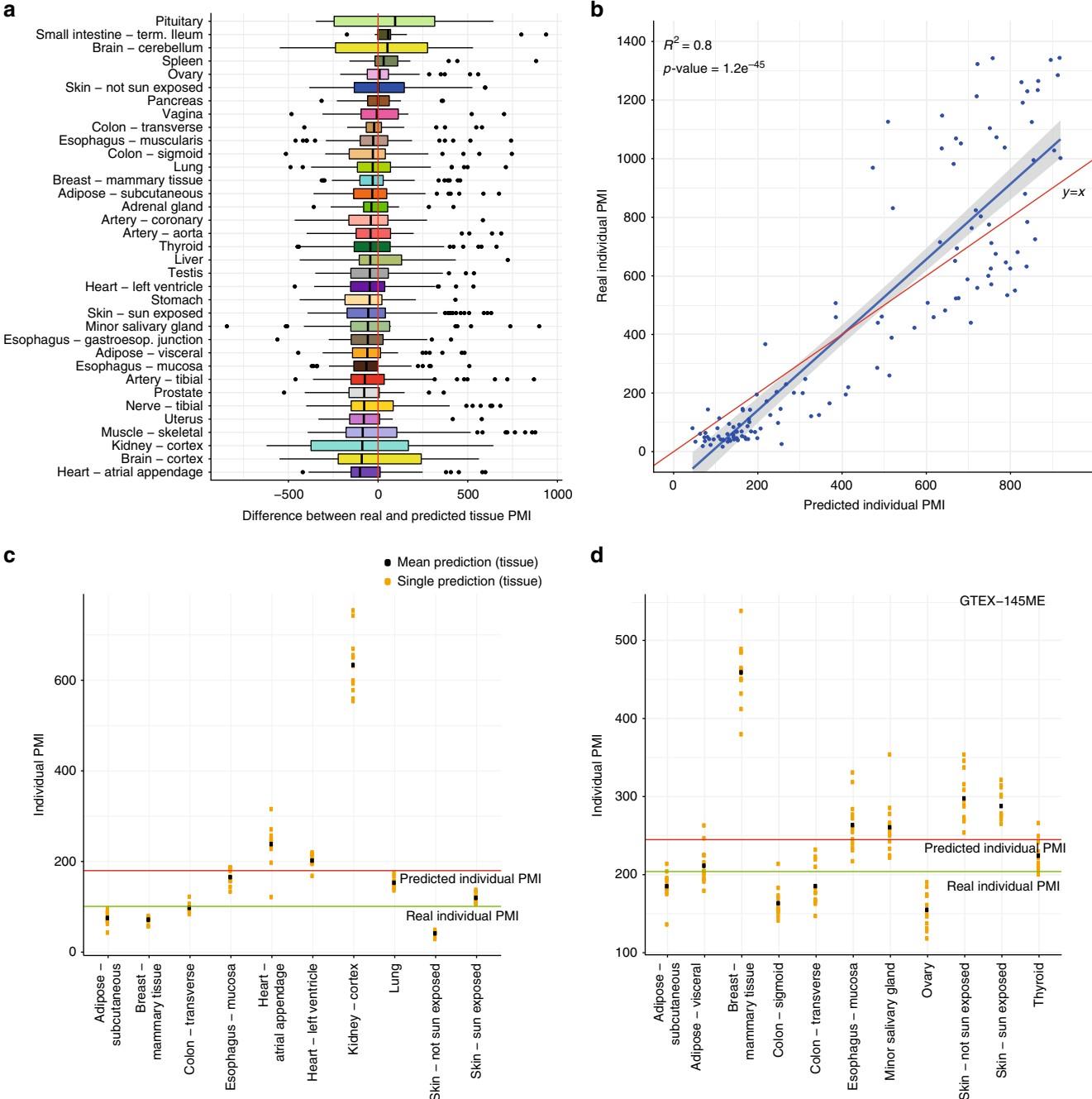

**Fig. 8** Prediction of the PMI from gene expression in post-mortem samples. **a** Distribution of the PMI prediction error per tissue. **b** Regression of the real PMI versus the predicted individual PMI on the test set of 129 individuals. Plots in panels (**c**) and (**d**) illustrate two examples (GTEX-145MN and GTEX-145ME) of the prediction of the PMI for an individual based on the prediction of PMI from each tissue from the individual. For a given tissue, each yellow dot represents a prediction from each one of 13 different models. The black dot is the mean prediction of these 13 models. The green line represents the real PMI value for the individual. The individual PMI prediction is calculated as the average of the final tissue PMI predictions, and is represented by the red line

enzymatic processes underlying mRNA degradation. However, our results suggest instead that there is ongoing regulation of transcription, at least during the hours immediately following death. We observed that in the majority of tissues there are many genes that display expression profiles that are more complex than simple monotonic changes with PMI. This is in agreement with a recent study by Pozhitkov et al.[29], in which gene expression profiles produced by cDNA microarrays were analyzed in zebrafish and mouse samples with post-mortem intervals up to 48 and 96 h. That study showed a non-monotonic increase in the

abundance of certain transcripts and suggested that previously silenced genes were actively transcribed at later post-mortem time points.

Similarly, analysis of splicing changes with PMI did not show conclusive evidence of systematic splicing deregulation (as measured by the splicing entropy) across tissues. Death, in contrast, did apparently lead to some splicing deregulation in blood. In particular, we found that the usage of the major isoform (the most abundant isoform compared to the rest) was attenuated in post-compared to pre-mortem samples. The usage of a major isoform

| Protocol for post-mortem interval prediction of an individual |
|---|
| 1. Annotate the start time of the forensic procedure, START_TIME. |
| 2. Annotate the stabilization time of the tissues, among the following ones, that are available: |
| o Skin sun exposed (lower leg) sample, SKNS_STAB. |
| o Adipose Subcutaneous sample, ADPSC_STAB. |
| o Thyroid sample, THYROID_STAB. |
| o Lung sample, LUNG_STAB. |
| 3. Prepare sample, extract RNA and perform RNA sequencing (RNA-seq) analysis, possibly with a portable and real-time equipment of the available tissues. |
| 4. Process the RNA-seq data to obtain gene expression quantifications for each available tissue sample. |
| 5. Provide to the PMI prediction software the gene expression values in each available tissue together with the stabilization time of the tissue, and the time of the initiation of the forensic procedure. |
| 6. The prediction software provides tissue and individual PMIs, as well as the coefficient of variation of the tissue PMIs as a measure of the stability of the predictions |

**Fig. 9** Protocol for post-mortem interval prediction. Steps to be performed to predict the PMI of an individual

across tissues and biological conditions has been reported as a general characteristic of genes[38,39], and has lead to extensive debate on the physiological relevance of regulated alternative splicing[39]. Since most data to date has been collected from post-mortem samples, the preferential usage of a single major isoform in living cells may be even more prevalent than previously reported.

While the effects of death per se on gene expression are distinct from those of increasing PMI, we found a number of genes involved in the assembly of DNA, nucleosome and chromatin that were affected by both. Expression changes of these genes suggest a possible form of gene regulation through the alteration of the chromatin structure. Pozhitkov et al.[29] described an increased expression of epigenetic regulatory genes, hypothesizing that the activation of these genes reveals the nucleosomes and allows for the later transcription of developmental genes that have no early expression. We consistently detected the upregulation of genes involved in DNA organization, but we could not detect changes in expression among genes related to development. This could be due to the lack of reference expression levels from very early post-mortem samples.

Based on the tissue specific response of the transcriptome to PMI, we built machine-learning models to predict the time of death of a recently deceased individual. We show that RNA-seq performed on a few key tissues could become a powerful tool to aid in forensic pathology. It could carry the footprint not only of the time since death, but also of the cause of death—even though we could not properly carry out these analyses because of the small sample sizes available. Interestingly, the most informative tissues to predict time of death included readily accessible ones, such as skin and subcutaneous adipose. While these results show promise, larger datasets more balanced across a wider post-mortem time interval will be required to assess the full potential of the approach.

In line with previous studies[12,13,15,17,20] our analyses show that the investigation of the impact of post-mortem ischemia in tissue transcriptomes is essential to properly interpret gene expression estimates obtained from post-mortem tissue samples. Furthermore, understanding the transcriptional changes occurring with time after death could have multiple applications. Here, we illustrated an application specific to forensic pathology, but other applications could include improving biospecimen procurement and organ preservation protocols. These could, in turn, have an impact on the procedures employed for organ transplantation.

Post-mortem tissues are irreplaceable sources for researching and understanding human biology, but as our study demonstrates, death does introduce a bias in the cellular transcriptomes, even over relatively short timeframes. Ideally cellular transcript levels should be measured "in vivo" in unperturbed cells to provide an unbiased characterization of true physiological cellular transcriptomes, and this should in turn be performed individually in each of the millions of cells that constitute a living tissue. However, current technologies for the genome-wide characterization of the transcriptome still require the dissociation and destruction of cells, even when obtained from living donors, and the impact of this cellular destruction on the transcriptome is also largely unknown. Despite this, our results indicate that overall, relatively few genes show significant changes over the post mortem intervals studied, and the genes that do change do not change systematically, but vary by cell and tissue type. To minimize and limit the impact of these changes, adhering to strict protocols and standards for the collection of high quality tissue and RNA, combined with careful documentation of all key sample procurement covariates (as was done for GTEx[21]), can allow the effects of post-mortem ischemia to be largely identified and corrected for in analyses. Post-mortem samples can therefore be of tremendous value for studies of both normal and disease biology.

## Methods

**Data and filters.** We used mRNA sequencing data (Illumina paired-end, 76 bp) from the GTEx project Analysis Freeze V6 release (phs000424.v6.p1). RNA-seq libraries are non-strand specific, with Poly-A selection and generated with Illumina TruSeq protocol. Further details on sample collection, processing and quality control of the RNA-seq samples from version V6 can be found in the supplementary material of[24] and in[21,23,40]. As described in Carithers et al.[21] the GTEx project made an effort to collect tissues within 8 h of PMI and RIN values ≥6. All samples under analysis in this study were collected and preserved with the PAXgene Tissue preservation system developed by Qiagen[21].

From the 55 available tissues in the V6, we started by selecting those with at least 20 samples. Brain samples are preserved either with PAXgene Preserved or Fresh Frozen methods. The latter does not have ischemic time available. We have included 161 samples from Cerebellum and 147 from Cortex preserved with PAXgene method and with ischemic time available. We further removed cell lines from the set of tissues. The final dataset comprises 36 tissues, with 36–1049 samples with a mean of 253 samples, see Supplementary Fig. 1.

**Gene expression.** RNA-seq reads were aligned to the human genome (hg19/GRCh37) using TopHat[41] (v1.4) and Gencode annotation v19[26] was used for gene quantification. We considered genes with at least 5 reads mapping in exons and from all biotypes in the annotation. The raw read counts were used for differential expression analysis and the RPKM[42] values, which were log2 transformed with an added pseudo-count used in the remaining analysis. For the regression analyses, the matrix of expression values was obtained for the samples of each tissue and then normalized with the normalize.quantiles function from the preprocessCore library[43].

In order to investigate the global patterns of gene expression we have considered the gene expression values of all the tissues and all the genes from the annotation[26]. We have then performed multi-dimensional scaling (MDS) using the isoMDS function from the package MASS in R. We defined the distance for two samples A and B as:

dist(A, B) = 1−PearsonCorrel(A, B). As shown in Supplementary Fig. 3 there is a clear transcriptional signature characterizing each tissue. Further discussion on detectable and tissue specificity expression and gene expression patterns across tissues can be found here[25].

**Post-mortem interval (PMI) information**. GTEx annotation provides information on three types of ischemic time: Total Ischemic time for a sample, Total Ischemic time for a donor and Ischemic Time (time for the start of the GTEx procedure). All these variables are quantified in minutes. Throughout the text we have used the term Post-Mortem Interval (PMI) to refer to ischemic time and except if explicitly stated it refers to the sample ischemic time. Negative PMI values (observed in blood samples) correspond to samples extracted pre-mortem. GTEx annotation contains a total of 249 sample and subject variables, which are divided in six groups: Sample attributes (prefix SM), Death circumstance (DTH), Demographic (no specific prefix), Medical History (MH), Tissue Recovery (TR), Serology results (LB). We selected those variables with $\geq 2$ and $\leq 15$ values, removing cases of unknown values. We then computed a linear regression with PMI, obtaining the adjusted $R^2$ and the Pearson correlation with PMI (cor.test in R with use = "na.or.complete"). Categorical variables were converted to numeric. Supplementary Fig. 4 shows the respective correlation values for all the variables, where we observe that TR, LB and DTH variables are highly correlated with ischemic time. This basically reflects different aspects of the tissue collection procedure that are highly associated with PMI.

**Covariate selection**. In order to assess the impact of PMI on gene expression we need to account for the possible effect of other pre-mortem variables on the variation of gene expression. For the selection of the variables of interest we excluded TR, LB and DTH variables due to their strong association with PMI, which may result from the fact that all these variables capture the underlying characteristics of death circumstances and tissue extirpation procedure. We then focused on sample, demographic and medical history variables (correlation values with ischemic time ignoring missing values), summing 166 variables. We further filter for those covariates that are qualitative and describe a phenotype such as age or gender and that exclude those that are simply metrics from the sequencing (SM). Finally, we kept those variables with $|r| > 0.1$ with PMI. Non-numerical variables are converted to numeric format. The final set of covariates used for regression analysis of PMI and gene expression is presented in Supplementary Table 5.

**Gene expression and PMI regression model**. In order to assess the impact of PMI on gene expression we took into account a set of fourteen covariates (Supplementary Table 5). Then, for each gene (with average expression across the tissue samples greater than 0.5 RPKM) we implemented a linear regression model where the gene expression profile is modeled with relation to the covariates: reg = lm (gene_expression ~ matrix.selectedCovariates). The residuals of the model are then used as the expression phenotype: gene_expression.resid = residuals(reg). Finally, the correlation between the residuals and the PMI is calculated, $r$ = correlation (gene_expression.resid, pmi.vals). The corresponding correlation and $p$-values (adjusted with BH method[44]) are then stored for all genes. This procedure is repeated for all tissues (Supplementary Fig. 37). To compute the correlation values of gene expression and PMI without the covariates, a procedure similar to the above was used where Pearson correlation is obtained between gene expression and PMI values (Supplementary Fig. 38). Supplementary Note 2 provides the algorithmic details of the methodology.

**Non-linear temporal differential expression**. In order to find non-linear differential expression we developed a method to identify significant changes between different post-mortem intervals. For each tissue we grouped the samples as in[21] and in five different PMI intervals I1: < 1 h, I2: ≥ 1 h and < 4 h, I3: ≥ 4 h and < 6 h, I4: ≥ 6 h and < 15 h and I5: ≥ 15 h. We then normalized the gene expression of each gene computing a Z-score = ((X−mean)/stdev) and calculated the median expression in each of these intervals. Every two consecutive intervals, with a minimum number of five samples are then compared. We consider an event of temporal differential expression between $T_i$ and $T_{i+1}$, where $T_i$ and $T_{i+1}$ correspond to the expression values of the gene in the interval $i$ and $i + 1$ if we meet the two following conditions:

$$\text{pval}(i, i+1) = \text{wilcox.test}(T_i, T_{i+1}), \text{with pval} < 0.05 \qquad (1)$$

$$\text{fold\_change}(i, i+1) = log_2(\text{median}(T_i)/\text{median}(T_{i+1})), |\text{fold\_change}(i, i+1)| > 2 \qquad (2)$$

Supplementary Fig. 39 provides the algorithmic details of the methodology.

**Tissue similarity for PMI correlated expression**. In order to build a tissue similarity matrix of the correlation profiles (gene expression and PMI) we performed a pairwise comparison of all tissues. For every pair of tissues we obtain the common genes by intersecting genes that in both tissues have correlation value of gene expression with PMI. For every pair of tissues we then obtain a Spearman ranking correlation based on the correlation values of the common genes. We then

used the heatmap.2 function from gplots to calculate the heatmap with dendrogram in Fig. 2f.

**Functional enrichment analysis**. For functional enrichment analysis we used the R libraries: DOSE[45], ClusterProfiler[46], Kegg.db[47] following the tutorial of ClusterProfiler[46].

**Differential expression in blood samples**. For differential expression analysis we used the statistical methods implemented in the edgeR package[48]. We started by building a matrix with gene read counts in premortem ($n = 169$) and postmortem ($n = 223$) Blood samples. Genes were filtered to have at least 5 reads per million mapped reads in at least 10% of the samples on one of the tested groups (cpm function). We created a design matrix taking into account 2 groups (pre- and postmortem samples) and several covariates:

design< − model.matrix(∼SMRIN + AGE + ETHNCTY + MHCANCERNM+ SMCENTER + SMTSTPTREF + SMNABTCHT + GENDER + group, covars.matrix)

Covariates SMRIN and AGE were discretized according to the following intervals: SMRIN = < 7*/7−8/8−9/9−10 and AGE = 20−30/30−40/40−50/50−60/60 −70. Covariates were converted as factors. See Supplementary Table 5 for the description of the covariates, where group variable corresponds to the pre and post-mortem samples. We then followed the protocol at[48] performing the normalization with the TMM method[49] Generalized Linear Model (GLM) based functions to estimate common dispersion and differential tests. For the differential expression analysis across different post-mortem blood intervals, we first divided the post-mortem samples in four groups, which provided an equal number of samples in each group: $G_1$ ($n = 56$): $0 < \text{pmi} \leq 406$ min; $G_2$ ($n = 56$): pmi > 406 and pmi $\leq 635$; $G_3$ ($n = 56$): pmi > 635 and pmi $\leq 867$; $G_4$ ($n = 55$): pmi > 867 and pmi $\leq 1401$. A similar approach as the one described above was then applied to compare all the pre-mortem samples with each of the $G_1$, $G_2$, $G_3$, and $G_4$ groups. Figure 4b and Supplementary Table 8 shows the number of differentially expressed genes for the different intervals.

**Transcriptional patterns of pre and post-mortem blood**. In order to explore the transcriptional differences in pre and post mortem blood samples we have built the respective expression matrix based on RPKM values that were then log2 converted and normalized with normalize.quantiles function as previously described. We then performed hierarchical clustering (HC) and multidimensional scaling (MDS). We defined the distance between samples a and b as, dist(a,b) = 1 − cor(a,b), where cor is the Pearson correlation of a and b vector. Hierarchical clustering solution was then computed with hclust function using the average method. Visualization was performed using the heatmap.2 function with the input of the distance matrix and the previously calculated HC solution as the dendrogram parameter. Postmortem samples ($n = 20$) with a PMI smaller than the respective individual PMI were excluded. Heatmap with PMI interval colors is shown in Supplementary Fig. 20, and samples in MDS plot were colored according to Hardy Scale (Fig. 4a).

**Signaling pathway models**. The hiPathia[34] tool was used for the interpretation of the consequences of the combined changes of gene expression levels and/or genomic mutations in the context of signaling pathways (see Supplementary Fig. 40 and 41). Significant circuits associated to PMI were obtained by fitting a linear model and were summarized by the median value across samples per circuit and time points. Supplementary Note 3 and Hidalgo et al.[34] provide the algorithmic details of the methodology.

**Gene structural features**. Features were derived from the Gencode annotation v19[26], including the number of projected (non-redundant exonic regions) exons, length of the coding regions, overall length of the gene, biotype. We obtained projected exons first by sorting by genomic coordinates and then by merging exons. We used bedtools[50] for this step. GC content was obtained from the Ensembl Biomart (www.ensembl.org/biomart). For each tissue we have calculated the Pearson correlation between the vector of gene features and the respective correlation value between gene expression and PMI. Supplementary Table 7 contains the correlation values per tissue for each feature.

**Mitochondrial transcription**. For estimating the mitochondrial RNA concentrations (MT%), we divided all reads in annotated mitochondrial (mt) genes by the total number of reads in annotated (nuclear and mitochondrial) genes. To account for the substantial different mitochondrial activity across tissues, we divided each sample by the median MT% found in the corresponding tissue (nMT%). We then regressed a linear model nMT% ~ PMI and compared the slopes obtained at each time point between the different tissues (Supplementary Fig. 17). Correcting for the influence of age in the linear model changed the distribution of relative MT% only marginally (shifted values $\leq 0.05$) (Supplementary Fig. 16).

**RNA-seq metrics across tissues**. We have explored if the different tissues show differences in RNA-seq quality control metrics obtained with the RNA-SeQC[40]

pipeline. Supplementary Table 11 lists the variables used for this analysis. The mapping proportions along the different gene features are shown in Supplementary Fig. 11 and 12. Degradation of the RNA may result in different mapping bias effects, in particular in a higher read coverage at the 3′ end of the genes. We have calculated for each sample a read coverage ratio between the 5′ and the 3′ 50bp-based normalization. Distribution of these values for the different tissues is shown in Supplementary Fig. 13 and the relation with RIN and PMI are shown in Supplementary Fig. 14 and 15.

**Clustering modularity.** To assess if gene expression signatures of tissues are preserved across the PMI bins, as defined above (section "Non-linear Temporal Differential Expression"), we selected only the tissues that had at least 10 samples within each bin of PMI. Because differences in the number of samples per tissues can introduce variation in the network structure, we randomly selected the same number of samples per tissue in each PMI bin, corresponding to the minimum number of samples per tissue across all the PMI bins. Thus, we have 4 combinations of 422 samples, one for each PMI bin, with the same tissues, and the same number of samples per tissue. For each combination of samples, we compute pairwise Pearson's correlation coefficient on the log2-transformed RPKM expression values after adding a pseudo-count of 1. From each matrix of correlation coefficients we built 7 networks, where nodes are the samples and edges are connections between samples that are correlated with a coefficient higher than a given threshold (out of 7 thresholds, from 0.86 to 0.92). These thresholds gave comparable network densities, defined as the proportion of connected nodes over the total number of possible edges, across all the networks. We used the modularity formula (R package igraph[51], modularity function) to measure how well the samples in each network are aggregated by tissue type. Supplementary Fig. 9 shows the distribution of modularity with relation to network density.

**Exon inclusion analysis.** GTEx samples were processed through the Integrative Pipeline for Splicing Analyses (IPSA) pipeline with default settings[52]. Namely, short reads were mapped to human genome (hg19/GRCh37) using TopHat[41] (v1.4). The alignments were filtered to have an overhang of at least 8 nt and entropy of the offset distribution of at least 1.5 bits. Novel short exons (shorter than the read length) were predicted using reads with more than one split with canonical GT/AG splicing nucleotides and minimum entropy of at least 1.5 bits for each splice junction. The percent-spliced-in (PSI) metric was computed as in Wang et al.[33] by using inclusion and exclusion reads with the minimum total count of 5 reads; that is, exons for which the combined number of inclusion and exclusion reads was less than 5 were excluded.

From the PSI values calculated by the IPSA pipeline[52] we have performed further filtering on a tissue basis based on the three following criteria: Select exons with NAs in less than 10% of the cases; Select exons they have a standard deviation greater than 0; Select exons if the difference between the max and min PSI values is larger than 0.1;

From this subset of selected exons we have then performed correlation analysis of the PSI value with the PMI value for each tissue. Supplementary Fig. 18a shows the number of tested exons according to the above filtering and Fig. 3d the number of significant exons at 1% FDR and |r| > 0.5.

**Differential exon inclusion in blood.** Exons with PSI values following the three criteria defined in the previous section in Blood samples were selected for differential expression analysis. This yielded a set of 3441 exons. Next, differential exon inclusion was tested with Wilcoxon Sum Rank test, with multiple testing adjustments by Benjamini-Hochberg method[44], and the median of the PSI values in each group calculated. Exons were deemed significant included if they pass the following criteria: FDR < 1%; and |ΔPSI| > 0.1;

**Cellular composition.** In order to perform gene expression signal deconvolution we applied CIBERSORT[35] v1.04 and the LM22 gene signature to all blood samples, using gene RPKMs, with default parameters, deconvoluting the signal into 22 different cell types. We discarded four cell types with average fraction below 0.01 in both conditions, keeping 18 cell types. We compared the cell-fractions of pre- and post-mortem samples globally using the Anderson–Darling test[53] and by cell-type obtaining a p-value using the two-sided Wilcoxon Rank-Sum test[54], adjusted to multiple-testing by Benjamini–Hochberg method[44].

**Splicing entropy analysis and PMI association.** To investigate changes in patterns of isoform usage and how these correlate with the PMI we have calculated the splicing entropy based on the relative abundance of an isoform/transcript within a gene. The following selection criteria and calculation was performed on a tissue-by-tissue basis: start by selecting genes with two or more isoforms; next, select genes with a non-zero expression in 90% of the samples of the tissue. Calculate isoform ratios for each gene: For a gene $G$, with $k$ isoforms $I$, the splicing ratio is defined as: $P(I_i) = \frac{I_i}{\sum_{i=1}^{k} I_i}$, where $I_i$ corresponds to the RPKM value for the isoform $i$ of $G$.

Finally, calculate the entropy of a gene based on the Shannon Entropy formula, as:

$$E(G) = -\sum_{i=1}^{k} p(I_i) \times \log p(I_i)$$

The splicing entropy of gene $G$ is maximal if all its isoforms have the same ratio and minimal if one of the isoforms dominates all the expression of $G$.

Then, for each gene, we correlate the splicing entropy with the respective PMI of the sample. From this test, we obtain the $r$-value and $p$-value. Perform $p$-value adjustment for multiple testing by Benjamini–Hochberg method[44]. We repeat this analysis for all the selected tissues. In Supplementary Table 10 we provide the total number of genes tested per tissues and the genes with a $|r| > 0.5$ and FDR < 5%. Figures 3g, h present an example of a gene with a significant change in lung. Supplementary Fig. 19 presents the distribution of the correlation values for Splicing Entropy and PMI across the different tissues.

**Machine learning models for PMI prediction.** The predictive model for PMI based on gene expression was constructed with a two-step approach using an ensemble of gradient boosted trees (Supplementary Fig. 24) in order to provide a robust estimate of PMI and avoid overfitting. 528 available individuals are initially partitioned into training and testing datasets, using 75% and 25% of the data, respectively. This partition is performed in such a way that we try to keep a similar underlying distribution of the number of available tissues per individual both for the training and testing datasets.

In order to build these tissue models (with the R implementation of the xgboost package[37]), we first create a fixed split of individuals into training and test sets. For a given tissue, we perform 3-repeat-5-fold cross validation with the samples corresponding to the individuals of the training block in order to select the best model, and we generate the predictions over the unseen test set using this model. This process is repeated 13 times using different seeds to take into account the variation in the hyperparameter optimization process. The output is a matrix of $n$ samples×13 columns, where each column represents the tissue PMI prediction of all test samples for each iteration. The final tissue PMI predictions will be taken as the row average of this matrix. Only protein coding genes with a correlation $>= 0.4$ with the tissue ischemic time were used in order to reduce the computational burden of model fitting. No other covariate was considered. Hyperparameter search was performed during this cross-validation loop using standard grid search for tree depth ranging from 4 to 6, η ranging from 0.001 to 0.1, γ ranging from 0 to 0.15 and up to 1000 rounds, using RMSE as optimization criteria. For each tissue we repeat the previous process 13 times using different seeds to determine the training set fold partitions in order to have a measure of the variability of the final prediction while applying the models on the test set (Supplementary Fig. 24b). On Supplementary Fig. 25a, we show the variability of the number of genes selected by the each 13 models, per tissue.

Once we have obtained the 13 models per tissue, we use each one of them to generate PMI predictions for each tissue sample in the test set. Therefore, 13 predictions are generated per tissue for a given test individual. We take the average of these predictions as our final PMI prediction for that particular tissue. On Supplementary Fig. 27 we can see examples of the tissue performance on the test set.

In the second step of our procedure, for each individual we will correct the final tissue PMI predictions by subtracting the elapsed time of the GTEx procedure. Since we know how much time has passed since the beginning of the GTEx procedure until a specific sample has been processed, this time difference has to be subtracted from the tissue PMI predictions in order to normalize them to a reference level, which is the start of the GTEx procedure. Finally, to predict the individual PMI (which is considered to be the time from death until the beginning of the procedure), we compute the average of these corrected tissue PMI predictions.

One important remark is that the final quality of the individual PMI prediction for the test individuals will highly depend on how accurate the individual tissue models are. For this reason, while performing the second step of the prediction procedure, we decided to use only the top 20 tissues with the best $R^2$ performance in the training data (Supplementary Fig. 25b). The $R^2$ for each of these tissues while applying the models on the test set is shown on Supplementary Fig. 26. The density of the individual PMI prediction error, which is defined as the signed difference of the real and predicted individual PMI is shown on Supplementary Fig. 30.

To investigate whether we are recovering a real predictive signal from gene expression instead of just an artifact, we used whole Blood samples, where there is information available for pre-mortem individuals as well. We reasoned if we were able to predict accurately the time to death for pre-mortem individuals this would reflect overfitting. To this end, for each cohort (pre-mortem and postmortem), we partitioned the data into training and testing datasets, fitted the model on the training data with 3-repeat-5-fold cross validation, performed the predictions on the test set and then obtained the regression statistics of real vs. predicted Blood PMI. We repeated this process a hundred times with different seeds, generating a different training/testing partition each time, in order to study the variability of the regression statistics. There is a significant difference of the regression statistics

between pre-mortem and postmortem samples, with a very poor fit for pre-mortem samples (Supplementary Fig. 29).

To study the stability of the tissue PMI predictive models we decided to evaluate the tissue performance irrespective of the training/testing partition used so far. For each tissue we partition all the available samples into 75% for training and 25% for testing. Once each tissue model is fitted on the training set with 3-repeat-5-fold cross validation, we calculate the $p$-value of the $F$-test (Supplementary Fig. 28), $R^2$ and slope (not shown) for the regression of the predicted tissue PMI in the test set versus the real tissue PMI. This process is then repeated 50 times by varying the training/testing partition in order to measure the variability of the regression statistics.

In order to find the optimal subsets of tissues for predicting an individual's PMI, we computed the individual PMI with all the possible combinations of sizes 2–6 of the available tissues for each individual in the test set. Then for each set size we keep the tissue combination that performed the best for each individual, in terms of individual PMI prediction error. With this, we can calculate the proportion of times a given tissue appeared in the optimal subset of a fixed size (Supplementary Fig. 32). We see that Adipose—Subcutaneous, Lung, Thyroid and Skin—Sun Exposed (Lower leg) are the tissues that consistently appeared on more optimal subsets of all sizes; and these tissues are also the ones that appear among the top most stable ones on the previously mentioned stability analysis. We then compute the individual PMI using all the possible combinations from size 2 to 4 using these four tissues (Supplementary Fig. 33 and 34).

As an additional description of stability of the individual PMI predictions, we have computed the standard deviation (SD) and coefficient of variation (CV) of the corrected tissue PMI predictions for each individual in our original test set, and generated the density curves of these statistics, shown on Supplementary Fig. 31, both for the top 20 tissues and the top 4 tissues of the best subset analysis.

Supplementary Fig. 35 shows the distribution of the death classes in the test set individuals of our PMI prediction methodology. In order to inspect if there is any immediate effect of the cause of death on the individual predictions, we have grouped the death classes in three larger categories: cerebrovascular disease, heart disease (which groups "Ischemic heart disease" and "Other forms of heart disease") and others (which groups the remaining classes). We observe that the performance of the predictions in terms of $R^2$ is very similar among the death classes.

To perform the prediction using TIN[15] data, we have employed the same methodology as with gene expression data, using the same initial partition of training and testing individuals. Since it is a proof of concept, we have only performed 3 repetitions of the process instead of 13 like in the gene expression method in order to reduce the computational burden. Supplementary Fig. 36a shows the performance of the model based on the TIN measure at the tissue level, while Supplementary Fig. 36b compares the predictions based on TIN with the predictions based on gene expression at the individual level. On Supplementary Fig. 36c, we show the number of most informative genes (defined as the union of genes with importance $>= 0.1$ across all the 13 models in the case of the gene expression model, and the union of the genes with importance $>= 0.1$ across the 3 models in the case of TIN, with "importance" being a measure computed by xgboost) with respect to each tissue and data type.

**Data Availability**. All data are available from dbGaP (accession phs000424.v6.p1) with multiple publicly available data views available from the GTEx Portal (www.gtexportal.org). The code can be obtained at https://public_docs.crg.es/rguigo/Papers/human_PMI_transcriptome/.

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

## Acknowledgements

We acknowledge and thank the donors and their families for their generous gifts of organ donation for transplantation and tissue donations for the GTEx research study. The Genotype-Tissue Expression (GTEx) project was supported by the Common Fund of the Office of the Director of the National Institutes of Health (commonfund.nih.gov/GTEx). We thank Anabela Nunes for help on image edition. This work was supported by the following grants and contracts: 1) From the US NIH: Contract HHSN261200800001E (Leidos Prime contract with NCI); Contracts 10XS170 (NDRI), 10XS171 (Roswell Park Cancer Institute), 10 × 172 (Science Care Inc.), and 12ST1039 (IDOX); Contract 10ST1035 (Van Andel Institute); Contract HHSN268201000029C (Broad Institute); R01 DA006227-17 (U Miami Brain Bank) 2) Ipatimup and i3s are partially funded by the Portuguese Foundation for Science and Technology (FCT); FEDER funds through the COMPETE 2020—Operacional Programme for Competitiveness and Internationalization (POCI), Portugal 2020, and by Portuguese funds through FCT/MCTES in the framework POCI-01-0145-FEDER-007274; 3) NORTE-01-0145-FEDER-000029, supported by NORTE 2020, under the PORTUGAL 2020 Partnership Agreement, through the European Regional Development Fund (ERDF); 4) FCT Fellowships SFRH/BPD/89764/2012 to PO, PD/BD/128007/2016 to AS; Salary support to PGF by POPH—QREN Type 4.2, European Social Fund and MCTES, program Investigador FCT, IF/01127/2014. 5) BIO2014-57291-R from the Spanish Ministry of Economy and Competitiveness and "Plataforma de Recursos Biomoleculares y Bioinformáticos" PT13/0001/0007 from the ISCIII, and EU H2020-INFRADEV-1-2015-1 ELIXIR-EXCELERATE (ref. 6,676559) Spanish Ministry of Economy and Competitiveness, 'Centro de Excelencia Severo Ochoa 2013-2017' and CERCA Programme/Generalitat de Catalunya, 7) Ministerio de Educación, Cultura y Deporte, under the FPU programme (Formación de Profesorado Universitario) with pre-doctoral fellowship FPU15/03635 to MMA. 8) Award Number 1R01MH101814 and 3R01MH101814-03S1 from the National Human Genome Research Institute. The content is solely the responsibility of the authors and does not necessarily represent the official views of the National Human Genome Research Institute or the National Institutes of Health. European Research Council under the European Union's Seventh Framework Programme (FP7/2007-2013) / ERC grant agreement RNA-MAPS_294653, Ministerio de Economía, Industria y Competitividad, referencia MEIC BIO2015-70777-P y los Fondos Estructurales de la Unión Europea (FEDER) y ISCIII del Ministerio de Economía y Competitividad, referencia PT13/0001/0021 cofinanciada por el Fondo Europeo de Desarrollo Regional (FEDER). 9) MS received funding from CNPq (grant 310132/2015-0 and 427541/2016-6), and from FAPERJ (grant E_06/2015).

## Author contributions

P.G.F. and R.G. conceived the study in coordination with K.A. M.M.A., F.R., and R.G. developed the PMI prediction models; C.P.S. and M.S. analyzed the mitochondrial transcription; A.B. the clustering modularity; A.A., M.R.H., J.C.C., C.C. and J.D. the signaling pathways; R.S. and D.D. the PSI calculation; J.C. and R.A. the signal deconvolution; A.S. contributed to the differential expression analysis; P.G.F. the remaining analysis. R.N., F.A., C.O., and P.O. contributed with suggestions to the analysis. P.G.F. and R.G. drafted the manuscript with contributions from M.M.A., F.R., J.D., M.S., and K.A. All authors revised the paper.

## Additional information

**Competing interests:** P.G.F., C.O., and P.O. are partners of Bioinf2Bio. The remaining authors declare no competing financial interest.

Pedro G. Ferreira [1,2], Manuel Muñoz-Aguirre[3,4,5,6], Ferran Reverter[3,4,5,7], Caio P. Sá Godinho[8], Abel Sousa[1,2,9], Alicia Amadoz[10], Reza Sodaei[3,4,5], Marta R. Hidalgo[11], Dmitri Pervouchine[3,4,5,12], Jose Carbonell-Caballero[13], Ramil Nurtdinov[3,4,5], Alessandra Breschi[3,4,5], Raziel Amador[3,4,5], Patrícia Oliveira[1,2], Cankut Çubuk [11], João Curado[3,4,5], François Aguet [14], Carla Oliveira[1,2], Joaquin Dopazo [10,11,15,16], Michael Sammeth [8], Kristin G. Ardlie[14] & Roderic Guigó[3,4,5]

[1]Instituto de Investigação e Inovação em Saúde, Universidade do Porto, Rua Alfredo Allen, 208, Porto, 4200-135, Portugal. [2]Institute of Molecular Pathology and Immunology, University of Porto, Rua Dr. Roberto Frias s/n, Porto, 4200-625, Portugal. [3]Centre for Genomic Regulation (CRG), The Barcelona Institute for Science and Technology, Dr. Aiguader 88, Barcelona, E-08003 Catalonia, Spain. [4]Universitat Pompeu Fabra (UPF), Barcelona, E-08003 Catalonia, Spain. [5]Institut Hospital del Mar d'Investigacions Mediques (IMIM), Barcelona, E-08003 Catalonia, Spain. [6]Departament d'Estadística i Investigació Operativa, Universitat Politècnica de Catalunya, Barcelona, E-08034 Catalonia, Spain. [7]Universitat de Barcelona, Barcelona, E-08028 Catalonia, Spain. [8]Institute of Biophysics Carlos Chagas Filho (IBCCF), Federal University of Rio de Janeiro (UFRJ), Rio de Janeiro, 21941-902, Brazil. [9]European Molecular Biology Laboratory, European Bioinformatics Institute (EMBL-EBI), Wellcome Genome Campus, Cambridge, CB10 1 SD, UK. [10]Department of Bioinformatics, Igenomix S.A, Valencia, 46980, Spain. [11]Clinical Bioinformatics Area,

Fundación Progreso y Salud (FPS), Hospital Virgen del Rocio, Sevilla, 41013, Spain. [12]Skolkovo Institute of Science and Technology, 100 Novaya Street, Skolkovo, Moscow Region 143025, Russia. [13]Chromatin and Gene expression Lab, Gene Regulation, Stem Cells and Cancer Program, Centre de Regulació Genòmica (CRG), The Barcelona Institute of Science and Technology, PRBB, Barcelona, 08003, Spain. [14]The Broad Institute of MIT and Harvard, Cambridge, MA 02142, USA. [15]Functional Genomics Node (INB), FPS, Hospital Virgen del Rocio, Sevilla, 41013, Spain. [16]Bioinformatics in Rare Diseases (BiER), Centro de Investigación Biomédica en Red de Enfermedades Raras (CIBERER), FPS, Hospital Virgen del Rocio, Sevilla, 41013, Spain. Manuel Muñoz-Aguirre and Ferran Reverter contributed equally to this work.

