## [Peer Review File · Nature Communications]

Reviewers' comments:

Reviewer #1 (Remarks to the Author):

The manuscript under review, "The effects of death and post-mortem cold ischemia on human tissue transcriptomes", submitted by Roderic Guigó et al., presents the results of a large and detailed study that was aimed at exploring the association between post-mortem tissue degradation and changes in RNA degradation / gene-expression. For this, they utilised the results of the GTEx project. More specifically, they used gene-expression data from 36 human tissues from 540 different donors.

One of the major aims for this manuscript was to describe – and here I quote the authors - "a model to predict the time since death from the analysis of the transcriptome of a few readily accessible tissues, describing for the first time an application of RNA-seq to forensic pathology".

In order to deliver this promise, the authors present in a clear and as concise as possible, considering the vast number of mainly descriptive statistical explorations.

I have two points of critique. First, I am surprised to see that, eventually, the authors rely on simple linear regression to report the relation between real and predicted PMI (see e.g. Fig 5, sFig 32). I am not convinced that there is a linear correlation, and this should be explored in much more detail.

Second, the most crucial aspect in any forensic trait prediction is the accuracy of the reconstructed prediction. There are many previous examples in the recent forensic literature where authors show not only how they derive a useful prediction model, but also describe how to infer the accuracy of the prediction, which is crucial. Again, this should be explored and described.

As a third, more minor point, I feel that the manuscript study would improve substantially if there would be a much more clear end-point: why not explain or describe in detail, the steps any forensic pathologist should or could take in order to be able to infer a reliable PMI, based on the information described in this manuscript. If, as the authors suggest, the RNA degradation profile of only four tissues is sufficiently reliable to estimate the PMI, a prototype protocol would make this a very important reference.

And finally, I am not so sure that this manuscript describes for the first time the relation between RNA-seq and PMI. More specifically reference 24 (from 2011) clearly predates this study (and there much more doing so). Also, references 8 and 12 cite the same article.

Reviewer #2 (Remarks to the Author):

The authors have analyzed GTEx gene expression data including 36 human tissues and donors each with at least 20 samples across different post-mortem intervals (PMIs). They found that many genes change expression over relatively short PMIs in a largely tissue-specific manner. The effect of this on biological interpretation and analysis is minimized. The most interesting results are that they identified the cascade of transcriptional events triggered by the death of the organism, where these events do not appear to simply reflect stochastic variations resulting from mRNA degradation, but active and ongoing regulation of transcription. In my opinion, this certainly provided a rich data resource for the community. However, the biological insights from the analysis seems very vague. The computational modeling lacks of novelty. For example, they should develop some sophisticated machine learning or clustering methods to examine or integrate all data sets together. The prediction model to predict PMI seems meaningless. Another noticeable weakness is that although authors list many factors on death, they didn't really focused on mining the data to correlate (or link) this expression levels with causes of deaths. This might be much useful for RNA-seq as a valuable tool in forensic pathology. There are some additional comments:

1. Are these data being analyzed in GTEx the 1st publication or reanalyzed?
2. Are they required to submit to GEO or ArrayExpress, some sorts of public data resources?

3. The summary of sequencing reads for each sample are missing.
4. Why does the Method include in Suppl. Material? It should move some part to the main text.
5. In Fig 4d, the words in pathways' nodes are very vague.
6. In Fig 5b, what is the p-value for the correlation?

Reviewer #3 (Remarks to the Author):

The manuscript

The effects of death and post-mortem cold ischemia 1 on human tissue transcriptomes
by Pedro G. Ferreira et al.

systematically analyses postmortem changes of the transcriptome of a large collection of tissues. The study has potentially high impact because practically all transcriptome studies use post-mortem samples although usually postmortem effects are ignored.

The study is well done and the main result states that postmortem effects have mostly no big disturbing effect on the outcome of expression studies. In my opinion alone this result is noteworthy.

The paper is exceptionally information-rich, well documented and (nevertheless) well written in a clear style. Almost all relevant information is provided in the supplementary files.

Taken together, the paper sheds new light into postmortem effects on tissue transcriptomes.

The only critical point from my side refers to the last subsection of the Results section: 'Prediction of the Post-mortem Interval from gene expression patterns 264 across multiple...'. I understood that the authors successfully applied machine learning models and they provide statistical data in support of this. I am not very common with the method and from a practical point of view I don't understand how to use these results to estimate, e.g. PMI given transcriptome data. Hence I miss a sort of practical approach in terms of a formulae, program-tool or whatever. Possibly the authors want to clarify this point.

For example, the authors show remarkable correlations between RIN and PMI on one hand and between PMI and the 3'/5' ratios. Why not to use e.g. the 3'/5' ratio as a rough estimate of PMI (possibly in a tissue specific way)?

Reviewers' comments:

Reviewer #1 (Remarks to the Author):

The manuscript under review, “The effects of death and post-mortem cold ischemia on human tissue transcriptomes”, submitted by Roderic Guigó et al., presents the results of a large and detailed study that was aimed at exploring the association between post-mortem tissue degradation and changes in RNA degradation / gene-expression. For this, they utilised the results of the GTEx project. More specifically, they used gene-expression data from 36 human tissues from 540 different donors.

One of the major aims for this manuscript was to describe – and here I quote the authors - “a model to predict the time since death from the analysis of the transcriptome of a few readily accessible tissues, describing for the first time an application of RNA-seq to forensic pathology”. In order to deliver this promise, the authors present in a clear and as concise as possible, considering the vast number of mainly descriptive statistical explorations.

I have two points of critique.

1. First, I am surprised to see that, eventually, the authors rely on simple linear regression to report the relation between real and predicted PMI (see e.g. Fig 5, sFig 32). I am not convinced that there is a linear correlation, and this should be explored in much more detail.

AR: We would like to clarify this point. We fully agree with the reviewer that there is no linear relationship between PMI and patterns of variation of gene expression. Thus, our predictive model for PMI based on gene expression was constructed using an ensemble of gradient boosted trees, which is a state-of-the-art machine learning non linear method to model both continuous and categorical responses. We constructed an ensemble of these models per tissue in order to provide a robust estimate of PMI and avoid overfitting as much as possible. For a given individual, we estimate the PMI (which is a single value) from the set of tissue PMI (which is as many values as tissues available).

Linear regression was only used to compare the predicted and the real PMI values, but not to predict PMI. When modeling a predictive outcome, R^2 is one of the most common statistics used to provide information about the performance of the model, as discussed on [<https://www.ncbi.nlm.nih.gov/pmc/articles/PMC3575184/pdf/nihms438237.pdf>, page 3]. If there is no linear correlation between predicted and real values, the predictive model is not accurate. As an additional metric of the accuracy of the predictions, we also provide the prediction error on Figure S30. Finally, we developed a stability analysis on which we inspect the variability of the performance of the models per tissue (Figure S31). Together, these analyses support the idea that there is predictive power on gene expression data to infer PMI.

6. In Fig 5b, what is the p-value for the correlation?

AR: In the revised version of the manuscript, we are adding on figure 5b the p-value ($1.25e-45$) for an association test using the Pearson product-moment correlation coefficient.

2. Second, the most crucial aspect in any forensic trait prediction is the accuracy of the reconstructed prediction. There are many previous examples in the recent forensic literature where authors show not only how they derive a useful prediction model, but also describe how to infer the accuracy of the prediction, which is crucial. Again, this should be explored and described.

AR: We thank the reviewer for this suggestion. The gradient-boosted trees that we have employed do not allow for a direct estimation of the reliability of the prediction, however, in an attempt to minimize uncertainty in predictions due to variations at model training time, we have fitted 13 different replications in order to stabilize the prediction value, as shown on the examples on Fig 5c and d. Moreover, since our final prediction of the PMI of an individual is based on the ensemble prediction of the PMI estimated on multiple tissues from the individual, we can use the consistency of these estimations as a measure of reliability. We reason that if all tissues predict consistently very similar PMI the prediction on the individual is more reliable than if the predictions are very variable across tissues. Thus, we propose to use the coefficient of variation (cv) as an indication of the reliability of the predictions--with lower values associated to more reliable predictions. We have included this on the revised version of our manuscript, including a supplementary figure (Figure S31) showing the distribution of cv of the predictions in the test set.

3. As a third, more minor point, I feel that the manuscript study would improve substantially if there would be a much more clear end-point: why not explain or describe in detail, the steps any forensic pathologist should or could take in order to be able to infer a reliable PMI, based on the information described in this manuscript. If, as the authors suggest, the RNA degradation profile of only four tissues is sufficiently reliable to estimate the PMI, a prototype protocol would make this a very important reference.

AR: We thank the reviewer for this very good suggestion, which is also given by reviewer #3. We have included a text box (Table 1), describing the necessary steps of our proposed protocol for PMI prediction in a real case forensic scenario.

4. And finally, I am not so sure that this manuscript describes for the first time the relation between RNA-seq and PMI. More specifically reference 24 (from 2011) clearly predates this study (and there much more doing so). Also, references 8 and 12 cite the same article.

AR: The reviewer is certainly right that there are some studies that precede our work on the prediction of PMI from the transcriptome. We believe that we acknowledge this by citing the appropriate references. Here, we meant the specific use of RNA-seq technology to assay the

transcriptome, which provides higher resolution on a larger number of genes than for instance microarrays, but also provides information on gene structure that is not easily available on other assays. The work of Koppelman et al (ref. 24) is indeed an important work but they approached a different question. Their goal was to evaluate the impact of several parameters, including PMI, on the integrity of the RNA. They used RT-PCR and RIN assays. From a characterization point of view, the work of Pozhitkov is the most similar to ours, but they focus on the transcriptome of mouse and zebrafish. From the PMI estimation point of view, the work of Sampaio-Silva is the most similar to ours, but they focus on a few genes in only three tissues. In both cases they use microarray assays. We have modified the text to clearly point out to these previous studies, which were all cited in our original submission, but we still think that our work is the first in which RNA-Seq is used to predict PMI.

Reviewer #2 (Remarks to the Author):

The authors have analyzed GTEx gene expression data including 36 human tissues and donors each with at least 20 samples across different post-mortem intervals (PMIs). They found that many genes change expression over relatively short PMIs in a largely tissue-specific manner. The effect of this on biological interpretation and analysis is minimized. The most interesting results are that they identified the cascade of transcriptional events triggered by the death of the organism, where these events do not appear to simply reflect stochastic variations resulting from mRNA degradation, but active and ongoing regulation of transcription. In my opinion, this certainly provided a rich data resource for the community. However, the biological insights from the analysis seems very vague.

1. The computational modeling lacks of novelty. For example, they should develop some sophisticated machine learning or clustering methods to examine or integrate all data sets together.

AR: We understand that the reviewer is referring to the computational model used to predict PMI. We are a little bit surprised by her/his comment that our “computational modeling lacks of novelty. For example, they should develop some sophisticated machine learning or clustering methods to examine or integrate all data sets together.” We do have developed a novel methodology using machine-learning methods. Specifically, we have developed an ensemble procedure based on gradient-boosted trees (which is a state-of-the-art standard machine learning method) in order to develop a proof-of-concept case study that shows that gene expression in post-mortem samples can be effectively used to predict time since death. As far as we know, this problem has never been attacked before and, consequently, no method like ours has ever been developed.

We understand that when the reviewer refers to the integration of multiple data sets simultaneously, he/she refers to the fact that we use a two step approach in which we first predict PMI for each tissue separately, and then, in the second step, from the the tissue PMIs we predict the PMI for a given individual. He/she may suggest that we should integrate data

from all tissues simultaneously to predict the individual PMI. The usage of data integration/fusion methods is non-trivial in practice in this particular case, due to several reasons, some of them are:

- a) the handling of missing data: developing a pipeline that takes data integration into account could mean that a prediction could be nullified or invalidated if data (gene expression) is not available for a given tissue
- b) data integration methods can be very computationally expensive: many of them rely on networks or bayesian computation. Many of the current methods become prohibitive when attempting to integrate many matrix layers (a matrix layer would be sample information for different individuals for a given tissue).
- c) Some data integration methods rely on latent spaces/dimensionality reduction and/or heavy regularization and therefore have loss of information when casting the data into a new feature space.

For these reasons, we believed it was more practical to explore the ensemble approach which we consider is more flexible: all the available data for a given individual is used without model-constraints requiring datasets of fixed sizes (i.e. availability of a samples throughout sets of tissues). Given the characteristics of the GTEx dataset (sample size and tissue sample availability) and for the purposes of this study, we decided to focus only on exploring and proving the potential of using RNA-seq as a forensic tool.

Having said this, the reviewer is right that data integration is a very interesting future avenue to explore: in particular, tensor regression is an approach that aims to use the data of several matrix layers in order to estimate a response, and it could be possibly applied for multiple-dataset problems such as this one. Thus, while we consider our work as groundbreaking by stating the problem clearly for the first time, and by providing a sensible solution, we expect that other developments could follow that will improve on our initial solution.

2. The prediction model to

predict PMI seems meaningless. Another noticeable weakness is that although authors list many factors on death, they didn't really focused on mining the data to correlate (or link) this expression levels with causes of deaths. This might be much useful for RNA-seq as a valuable tool in forensic pathology.

AR: We do not understand why the reviewer considers our "model to predict PMI meaningless." We believe that we show that within the time frame of GTEx PMI, our predictions are quite accurate. We agree that predicting the cause of death is also a very useful in forensic pathology, and we thank the reviewer for the suggestion. We actually mentioned this in our discussion, but we probably we did not elaborate enough. The plot next shows the distribution of the frequency of causes of death in the GTEx test set that we used in our submitted analysis.

As it is possible to see, for most death causes the sample size is too small for any meaningful statistical analysis. Still, following the reviewer's suggestion, we have attempted to perform some prototype analysis by grouping all causes related to heart disease, and creating three major death classes: cerebrovascular diseases, heart diseases and others. We used an *L2-regularized Support Vector Machine* to build a model for each tissue. We $\log_2(x+1)$ transformed the data, removing genes with near zero variance, and then centering and scaling both training and test datasets according to the values corresponding to the training set. We fit the model using 3-repeat-5-fold cross validation. We use a voting schema to generate the final cause of death prediction for a given individual, by choosing the most frequent prediction across all tissue predictions for the individual. The table next shows the accuracy of the predictions.

```
## Confusion Matrix and Statistics
##
##                            Reference
## Prediction                  Cerebrovascular_diseases Heart_disease Other
##   Cerebrovascular_diseases                        37            10    27
##   Heart_disease                                    2            30    28
##   Other                                            4             5    14
##
## Overall Statistics
##
##                Accuracy : 0.5159
##                  95% CI : (0.4349, 0.5963)
##     No Information Rate : 0.4395
##     P-Value [Acc > NIR] : 0.03261
```

```

##
##           Kappa : 0.3055
## McNemar's Test P-Value : 2.294e-08
##
## Statistics by Class:
##
##           Class: Cerebrovascular_diseases Class: Heart_disease
## Sensitivity           0.8605           0.6667
## Specificity           0.6754           0.7321
## Pos Pred Value        0.5000           0.5000
## Neg Pred Value        0.9277           0.8454
## Prevalence            0.2739           0.2866
## Detection Rate        0.2357           0.1911
## Detection Prevalence  0.4713           0.3822
## Balanced Accuracy     0.7680           0.6994
##
##           Class: Other
## Sensitivity           0.20290
## Specificity           0.89773
## Pos Pred Value        0.60870
## Neg Pred Value        0.58955
## Prevalence            0.43949
## Detection Rate        0.08917
## Detection Prevalence  0.14650
## Balanced Accuracy     0.55031

```

While the overall accuracy is poor, we believe that this is mostly because of the predictions in the class corresponding to the other causes of death. This is expected, being this class highly heterogeneous. Actually, the predictions for the heart and cerebrovascular diseases are reasonably accurate. Thus, we believe that while the results are promising, they are too premature to be included in a manuscript. Larger samples sizes are required to make proper predictions for cause of death. We have extended some the discussion to refer to this possibility.

We do have however, investigated whether cause of death has an impact on the accuracy of the PMI predictions. The next figure shows the R^2 of the regression between real and predicted PMI on the test set individuals for the three death classes above.

The accuracy remains similar in the three classes, suggesting that the cause of death does not have an important impact on PMI predictions. We have included these results in the manuscript (as well as the barplot with the frequency of cause of death in the test set samples, see Supplementary Figure 34).

There are some additional comments:

1. Are these data being analyzed in GTEx the 1st publication or reanalyzed?

AR: These results represent a first time analysis of version 6 of the GTEx dataset (8555 samples, 544 individuals), 4.3 times more samples than reported in the GTEx Pilot Phase. Previous papers published by consortium on the pilot phase refer to version 4 of the dataset (1641 samples). The GTEx analysis of the version of 6 have just been published (GTEx Consortium, Genetic effects on gene expression across human tissues, Nature 550, 2017).

2. Are they required to submit to GEO or ArrayExpress, some sorts of public data resources?

AR: All the data used in this analysis is freely available to the research community through the gtex portal, <https://gtexportal.org/home/datasets> and through dbGap portal (dbGaP Accession phs000424.v6.p1). Protected and raw data can be applied through dbGap portal. We have now added a data availability section to the main manuscript.

3. The summary of sequencing reads for each sample are missing.

AR: We have now added a table in supplementary materials (Supplementary Data) with the statistics (mean and standard deviation) for the total number of mapped reads and the mapping Rate (ratio of total mapped reads to total reads). Additional details on the QC of the RNA-seq samples are provided in the supplementary material of this manuscript and the consortium

manuscript “Genetic effects on gene expression across human tissues”, Nature 550, 204-2013, 2017.

4. Why does the Method include in Suppl. Material? It should move some part to the main text.

AR: We have now moved to the main paper the most of the Methods. Supplementary methods provide more algorithmic details.

5. In Fig 4d, the words in pathways' nodes are very vague.

AR: This correspond to the GO terms as used by the hiPathia tool. We have now added further details on the explanation of the results of figure 4d.

6. In Fig 5b, what is the p-value for the correlation?

AR: The p-value for the test for association/correlation between predicted vs. real PMI using Pearson's correlation coefficient (with H0: true correlation is equal to 0 and H1: true correlation is not equal to 0) is 1.2e-45. We have now added this information to the figure.

Reviewer #3 (Remarks to the Author):

The manuscript The effects of death and post-mortem cold ischemia on human tissue transcriptomes by Pedro G. Ferreira et al. systematically analyses postmortem changes of the transcriptome of a large collection of tissues.

The study has potentially high impact because practically all transcriptome studies use post-mortem samples although usually postmortem effects are ignored.

The study is well done and the main result states that postmortem effects have mostly no big disturbing effect on the outcome of expression studies. In my opinion alone this result is noteworthy.

The paper is exceptionally information-rich, well documented and (nevertheless) well written in a clear style. Almost all relevant information is provided in the supplementary files.

Taken together, the paper sheds new light into postmortem effects on tissue transcriptomes.

AR: We are glad that the reviewer considers our work of importance.

1. The only critical point from my side refers to the last subsection of the Results section: 'Prediction of the Post-mortem Interval from gene expression patterns 264 across multiple...'. I understood that the authors successfully applied machine learning models and they provide statistical data in support of this. I am not very common with the method and from a practical point of view I don't understand how to use these results the estimate, e.g. PMI given transcriptome data. Hence I miss a sort of practical approach in terms of a formulae, program-tool or whatever. Possibly the authors want to clarify this point.

AR: We thank the reviewer for this suggestion that has also been made by reviewer #1. We have included a text box (Table 1), describing all the steps required to apply our proposed approach we have included Table 1 in the main text describing the necessary steps of our proposed the protocol for PMI prediction.

2. For example, the authors show remarkable correlations between RIN and PMI on one hand and between PMI and the 3'/5' ratios. Why not to use e.g. the 3'/5' ratio as a rough estimate of PMI (possibly in a tissue specific way)?

AR: We thank the reviewer for this suggestion. Rather than the 3'/5' ratio, however, we have used the Transcript Integrity Number (TIN) which has been recently introduced and shown to correlate with RIN and RNA degradation (Wang et al, BMC Bioinformatics, 17:58, 2016). TIN measures the uniformity of the read distribution across the transcript length. We have employed exactly the same procedure to train a predictive model based on TIN values that the one we employed with gene expression values. Predictions based on TIN have accuracy which is very similar to that of predictions based on gene expression. Interestingly, the set of genes that are employed in the two models are largely divergent (see Figure S36). This is consistent with our finding that post-mortem transcriptomic changes are both the result of RNA degradation and of regulated gene expression, and opens the possibility of increasing the accuracy of the predictions by combining gene expression and RNA degradation values. We have discussed this in the revised version of the manuscript in which we have also included the prediction results based on TIN, including an additional Supplementary Figure 36.

REVIEWERS' COMMENTS:

Reviewer #1 (Remarks to the Author):

This reviewer has no further comments

Reviewer #2 (Remarks to the Author):

The authors have addressed my concerns.

Reviewer #3 (Remarks to the Author):

no comments